# TABDIFF: A MIXED-TYPE DIFFUSION MODEL FOR TABULAR DATA GENERATION

**Juntong Shi**[2][†], **Minkai Xu**[1][†][*] **Harper Hua**[1][†], **Hengrui Zhang**[3][†],
**Stefano Ermon**[1], **Jure Leskovec**[1]
[1]Stanford University [2]University of Southern California [3]University of Illinois Chicago

## ABSTRACT

Synthesizing high-quality tabular data is an important topic in many data science tasks, ranging from dataset augmentation to privacy protection. However, developing expressive generative models for tabular data is challenging due to its inherent heterogeneous data types, complex inter-correlations, and intricate column-wise distributions. In this paper, we introduce TABDIFF, a joint diffusion framework that models all mixed-type distributions of tabular data in one model. Our key innovation is the development of a joint continuous-time diffusion process for numerical and categorical data, where we propose feature-wise learnable diffusion processes to counter the high disparity of different feature distributions. TABDIFF is parameterized by a transformer handling different input types, and the entire framework can be efficiently optimized in an end-to-end fashion. We further introduce a mixed-type stochastic sampler to automatically correct the accumulated decoding error during sampling, and propose classifier-free guidance for conditional missing column value imputation. Comprehensive experiments on seven datasets demonstrate that TABDIFF achieves superior average performance over existing competitive baselines across all eight metrics, with up to $22.5\%$ improvement over the state-of-the-art model on pair-wise column correlation estimations. Code is available at https://github.com/MinkaiXu/TabDiff.

## 1 INTRODUCTION

Tabular data is ubiquitous in various databases, and developing effective generative models for it is a fundamental problem in many data processing and analysis tasks, ranging from training data augmentation (Fonseca & Bacao, 2023), data privacy protection (Assefa et al., 2021; Hernandez et al., 2022), to missing value imputation (You et al., 2020; Zheng & Charoenphakdee, 2022). With versatile synthetic tabular data that share the same format and statistical properties as the existing dataset, we are able to completely replace real data in a workflow or supplement the data to enhance its utility, which makes it easier to share and use. The capability of anonymizing data and enlarging sample size without compromising the overall data quality enables it to revolutionize the field of data science. Unlike image data, which comprises pure continuous pixel values with local spatial correlations, or text data, which comprises tokens that share the same dictionary space, tabular data features have much more complex and varied distributions (Xu et al., 2019; Borisov et al., 2023), making it challenging to learn joint probabilities across multiple columns. More specifically, such inherent heterogeneity leads to obstacles from two aspects: 1) typical tabular data often contains mixed-type data types, *i.e.*, continuous (*e.g.*, numerical features) and discrete (*e.g.*, categorical features) variables; 2) within the same feature type, features do not share the exact same data property because of the different meaning they represent, resulting in different column-wise marginal distributions (even after normalizing them into same value ranges).

In recent years, numerous deep generative models have been proposed for tabular data generation with autoregressive models (Borisov et al., 2023), VAEs (Liu et al., 2023), and GANs (Xu et al., 2019) in the past few years. Though they have notably improved the generation quality compared to traditional machine learning generation techniques such as resampling (Chawla et al., 2002), the generated data quality is still far from satisfactory due to limited model capacity. Recently, with the rapid progress in diffusion models (Song & Ermon, 2019; Ho et al., 2020; Rombach et al., 2022),

---

*Corresponding author. ✉ minkai@cs.stanford.edu. [†]Equal contribution

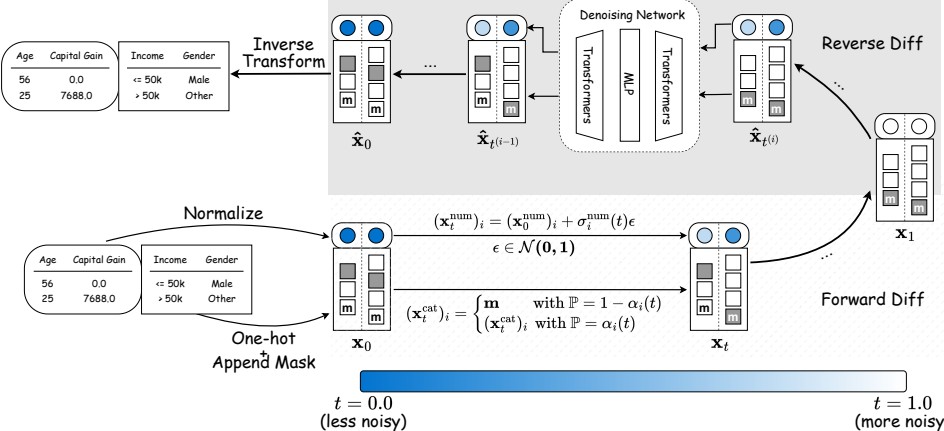

Figure 1: A high-level overview of TABDIFF. TABDIFF operates by normalizing numerical columns and converting categorical columns into one-hot vectors with an extra [MASK] class. Joint forward diffusion processes are applied to all modalities with each column's noise rate controlled by learnable schedules. New samples are generated via reverse process, with the denoising network gradually denoising $\mathbf{x}_1$ into $\mathbf{x}_0$ and then applying the inverse transform to recover the original format.

researchers have been actively exploring extending this powerful framework to tabular data (Kim et al., 2022; Kotelnikov et al., 2023; Zhang et al., 2024). For example, Zheng & Charoenphakdee (2022); Zhang et al. (2024) transform all features into a latent continuous space via various encoding techniques and apply Gaussian diffusion there, while Kotelnikov et al. (2023); Lee et al. (2023) combine discrete-time continuous and discrete diffusion processes (Austin et al., 2021) to deal with numerical and categorical features separately. However, prior methods are trapped in sub-optimal performance due to additional encoding overhead or imperfect discrete-time diffusion modeling, and none of them consider the feature-wise distribution heterogeneity issue in a mixed-type framework.

In this paper, we present TABDIFF, a novel and principled mixed-type diffusion framework for tabular data generation. TABDIFF perturbs numerical and categorical features with a joint diffusion process, and learns a single model to simultaneously denoising all modalities. Our key innovation is the development of mixed-type feature-wise learnable diffusion processes to counteract the high heterogeneity across different feature distributions. Such feature-specific learnable noise schedules enable the model to optimally allocate the model capacity to different features in the training phase. Besides, it encourages the model to capture the inherent correlations during sampling since the model can denoise different features in a flexible order based on the learned schedule. We parameterize TABDIFF with a transformer operating on different input types and optimize the entire framework efficiently in an end-to-end fashion. The framework is trained with a continuous-time limit of evidence lower bound. To reduce the decoding error during denoising sampling, we design a mixed-type stochastic sampler that automatically corrects the accumulated decoding error during sampling. In addition, we highlight that TABDIFF can also be applied to conditional generation tasks such as missing column imputation, and we further introduce classifier-free guidance technique to improve the conditional generation quality.

TABDIFF enjoys several notable advantages: 1) our model learns the joint distribution in the original data space with an expressive continuous-time diffusion framework; 2) the framework is sensitive to varying feature marginal distribution and can adaptively reason about feature-specific information and pair-wise correlations. We conduct comprehensive experiments to evaluate TABDIFF against state-of-the-art methods across seven widely adopted tabular synthesis benchmarks. Results show that TABDIFF consistently outperforms previous methods over eight distinct evaluation metrics, with up to 22.5% improvement over the state-of-the-art model on pair-wise column correlation estimations, suggesting our superior generative capacity on mixed-type tabular data.

## 2 METHOD

### 2.1 OVERVIEW

**Notation**. For a given mixed-type tabular dataset $\mathcal{T}$, we denote the number of numerical features and categorical features as $M_{\text{num}}$ and $M_{\text{cat}}$, respectively. The dataset is represented as a collection of

data entries $\mathcal{T} = \{\mathbf{x}\} = \{[\mathbf{x}^{\mathrm{num}}, \mathbf{x}^{\mathrm{cat}}]\}$, where each entry $\mathbf{x}$ is a concatenated vector consisting of its numerical features $\mathbf{x}^{\mathrm{num}}$ and categorical features $\mathbf{x}^{\mathrm{cat}}$. We represent the numerical features as a $M_{\mathrm{num}}$ dimensional vector $\mathbf{x}^{\mathrm{num}} \in \mathbb{R}^{M_{\mathrm{num}}}$ and denote the $i$-th feature as $(\mathbf{x}^{\mathrm{num}})_i \in \mathbb{R}$. We represent each categorical column with $C_j$ finite categories as a one-hot column vector $(\mathbf{x}^{\mathrm{cat}})_j \in \{0,1\}^{(C_j+1)}$, with an extra dimension dedicated to the [MASK] state. The $(C_j + 1)$-th category corresponds to the special [MASK] state and we use $\mathbf{m} \in \{0,1\}^K$ as the one-hot vector for it. In addition, we define $\mathrm{cat}(\cdot; \boldsymbol{\pi})$ as the categorical distribution over $K$ classes with probabilities given by $\boldsymbol{\pi} \in \Delta^K$, where $\Delta^K$ is the $K$-simplex.

Different from common data types such as images and text, developing generative models for tabular data is challenging as the distribution is determined by mixed-type data. We therefore propose TABDIFF, a unified generative model for modeling the joint distribution $p(\mathbf{x})$ using a continuous-time diffusion framework. TABDIFF can learn the distribution from finite samples and generate faithful, diverse, and novel samples unconditionally. We provide a high-level overview in Figure 1, which includes a forward *diffusion* process and a reverse *generative* process, both defined in continuous time. The diffusion process gradually adds noise to data, and the generative process learns to recover the data from prior noise distribution with neural networks parameterized by $\theta$. In the following sections, we elaborate on how we develop the unified diffusion framework with learnable noise schedules and perform training and sampling in practice.

## 2.2 MIXED-TYPE DIFFUSION FRAMEWORK

Diffusion models (Sohl-Dickstein et al., 2015; Song & Ermon, 2019; Ho et al., 2020) are likelihood-based generative models that learn the data distribution via forward and reverse Markov processes. Our goal is to develop a principled diffusion model for generating mixed-type tabular data that faithfully mimics the statistical distribution of the real dataset. Our framework TABDIFF is designed to directly operate on the data space and naturally handle each tabular column in its built-in datatype. TABDIFF is built on a hybrid forward process that gradually injects noise to numerical and categorical data types separately with different diffusion schedules $\boldsymbol{\sigma}^{\mathrm{num}}$ and $\boldsymbol{\sigma}^{\mathrm{cat}}$. Let $\{\mathbf{x}_t : t \sim [0,1]\}$ denote a sequence of data in the diffusion process indexed by a continuous time variable $t \in [0,1]$, where $\mathbf{x}_0 \sim p_0$ are *i.i.d.* samples from real data distribution and $\mathbf{x}_1 \sim p_1$ are pure noise from prior distribution. The hybrid forward diffusion process can be then represented as:

$$q(\mathbf{x}_t \mid \mathbf{x}_0) = q\left(\mathbf{x}_t^{\mathrm{num}} \mid \mathbf{x}_0^{\mathrm{num}}, \boldsymbol{\sigma}^{\mathrm{num}}(t)\right) \cdot q\left(\mathbf{x}_t^{\mathrm{cat}} \mid \mathbf{x}_0^{\mathrm{cat}}, \boldsymbol{\sigma}^{\mathrm{cat}}(t)\right). \tag{1}$$

Then the true reverse process can be represented as the joint posterior:

$$q(\mathbf{x}_s \mid \mathbf{x}_t, \mathbf{x}_0) = q(\mathbf{x}_s^{\mathrm{num}} \mid \mathbf{x}_t, \mathbf{x}_0) \cdot q(\mathbf{x}_s^{\mathrm{cat}} \mid \mathbf{x}_t, \mathbf{x}_0), \tag{2}$$

where $s$ and $t$ are two arbitrary timesteps that $0 < s < t < 1$. We aim to learn a denoising model $p_\theta(\mathbf{x}_s | \mathbf{x}_t)$ to match the true posterior. In the following, we discuss the detailed formulations of diffusion processes for continuous and categorical features in separate manners. To enhance clarity, we omit the superscripts on $\mathbf{x}^{\mathrm{num}}$ and $\mathbf{x}^{\mathrm{cat}}$ when the inclusion is unnecessary for understanding.

**Gaussian Diffusion for Numerical Features**. In this paper, we model the forward diffusion for continuous features $\mathbf{x}^{\mathrm{num}}$ as a stochastic differential equation (SDE) $\mathrm{d}\mathbf{x} = \mathbf{f}(\mathbf{x}, t)\mathrm{d}t + g(t)\mathrm{d}\mathbf{w}$, with $\mathbf{f}(\cdot, t) : \mathbb{R}^{M_{\mathrm{num}}} \to \mathbb{R}^{M_{\mathrm{num}}}$ being the drift coefficient, $g(\cdot) : \mathbb{R} \to \mathbb{R}$ being the diffusion coefficient, and $\boldsymbol{w}$ being the standard Wiener process (Song et al., 2021; Karras et al., 2022). The revere generation process solves the probability flow ordinary differential equation (ODE) $\mathrm{d}\mathbf{x} = \left[\mathbf{f}(\mathbf{x}, t) - \frac{1}{2}g(t)^2 \nabla_\mathbf{x} \log p_t(\mathbf{x})\right]\mathrm{d}t$, where $\nabla_\mathbf{x} \log p_t(\mathbf{x})$ is the score function of $p_t(\mathbf{x})$. In this paper, we use the Variance Exploding formulation with $\mathbf{f}(\cdot, t) = \mathbf{0}$ and $g(t) = \sqrt{2[\frac{d}{dt}\boldsymbol{\sigma}^{\mathrm{num}}(t)]\boldsymbol{\sigma}^{\mathrm{num}}(t)}$, which yields the forward process :

$$\mathbf{x}_t^{\mathrm{num}} = \mathbf{x}_0^{\mathrm{num}} + \boldsymbol{\sigma}^{\mathrm{num}}(t)\boldsymbol{\epsilon}, \quad \boldsymbol{\epsilon} \sim \mathcal{N}(\mathbf{0}, \boldsymbol{I}_{M_{\mathrm{num}}}). \tag{3}$$

And the reversal can then be formulated accordingly as:

$$\mathrm{d}\mathbf{x}^{\mathrm{num}} = -[\frac{\mathrm{d}}{\mathrm{dt}}\boldsymbol{\sigma}^{\mathrm{num}}(\mathrm{t})]\boldsymbol{\sigma}^{\mathrm{num}}(\mathrm{t})\nabla_\mathbf{x} \log p_t(\mathbf{x}^{\mathrm{num}})\mathrm{dt}. \tag{4}$$

In TABDIFF, we train the diffusion model $\boldsymbol{\mu}_\theta$ to jointly denoise the numerical and categorical features. We use $\boldsymbol{\mu}_\theta^{\mathrm{num}}$ to denote numerical part of the denoising model output, and train the model via optimizing the denoising loss:

$$\mathcal{L}_{\mathrm{num}}(\theta, \rho) = \mathbb{E}_{\mathbf{x}_0 \sim p(\mathbf{x}_0)}\mathbb{E}_{t \sim U[0,1]}\mathbb{E}_{\boldsymbol{\epsilon} \sim \mathcal{N}(\mathbf{0}, \boldsymbol{I})} \left\| \boldsymbol{\mu}_\theta^{\mathrm{num}}(\mathbf{x}_t, t) - \boldsymbol{\epsilon} \right\|_2^2. \tag{5}$$

**Masked Diffusion for Categorical Features**, For categorical features, we take inspiration from the recent progress on discrete state-space diffusion for language modeling (Austin et al., 2021; Shi et al., 2024; Sahoo et al., 2024). The forward diffusion process is defined as a masking (absorbing) process that smoothly interpolates between the data distribution $\mathrm{cat}(\cdot; \mathbf{x})$ and the target distribution $\mathrm{cat}(\cdot; \mathbf{m})$, where all probability mass are assigned on the [MASK] state:

$$q(\mathbf{x}_t|\mathbf{x}_0) = \mathrm{cat}(\mathbf{x}_t; \alpha_t \mathbf{x}_0 + (1 - \alpha_t)\mathbf{m}). \tag{6}$$

$\alpha_t \in [0, 1]$ is a strictly decreasing function of $t$, with $\alpha_0 \approx 1$ and $\alpha_1 \approx 0$. It represents the probability for the real data $\mathbf{x}_0$ to be masked at time step $t$. By the time $t = 1$, all inputs are masked with probability 1. In practice this schedule is parameterized by $\alpha_t = \exp(-\boldsymbol{\sigma}^{\mathrm{cat}}(t))$, where $\boldsymbol{\sigma}^{\mathrm{cat}}(t) : [0, 1] \to \mathbb{R}^+$ is a strictly increasing function. Such forward process entails the step transition probabilities $q(\mathbf{x}_t|\mathbf{x}_s) = \mathrm{cat}(\mathbf{x}_t; \alpha_{t|s}\mathbf{x}_s + (1 - \alpha_{t|s})\mathbf{m})$, where $\alpha_{t|s} = \alpha_t/\alpha_s$. Under the hood, this transition means that at each diffusion step, the data will be perturbed to the [MASK] state with a probability of $(1 - \alpha_{t|s})$, and remains there until $t = 1$ if perturbed.

Similar to numerical features, in the reverse denoising process for categorical ones, the diffusion model $\boldsymbol{\mu}_\theta$ aims to progressively unmask each column from the 'masked' state. The true posterior distribution conditioned on $\mathbf{x}_0$ has the close form of:

$$q(\mathbf{x}_s|\mathbf{x}_t, \mathbf{x}_0) = \begin{cases} \mathrm{cat}(\mathbf{x}_s; \mathbf{x}_t) & \mathbf{x}_t \neq \mathbf{m}, \\ \mathrm{cat}\left(\mathbf{x}_s; \frac{(1-\alpha_s)\mathbf{m}+(\alpha_s-\alpha_t)\mathbf{x}_0}{1-\alpha_t}\right) & \mathbf{x}_t = \mathbf{m}. \end{cases} \tag{7}$$

We introduce the denoising network $\mu_\theta^{\mathrm{cat}}(\mathbf{x}_t, t) : C \times [0, 1] \to \Delta^C$ to estimate $\mathbf{x}_0$, through which we can approximate the unknown true posterior as:

$$p_\theta(\mathbf{x}_s^{\mathrm{cat}}|\mathbf{x}_t^{\mathrm{cat}}) = \begin{cases} \mathrm{cat}(\mathbf{x}_s^{\mathrm{cat}}; \mathbf{x}_t^{\mathrm{cat}}) & \mathbf{x}_t^{\mathrm{cat}} \neq \mathbf{m}, \\ \mathrm{cat}\left(\mathbf{x}_s^{\mathrm{cat}}; \frac{(1-\alpha_s)\mathbf{m}+(\alpha_s-\alpha_t)\boldsymbol{\mu}_\theta^{\mathrm{cat}}(\mathbf{x}_t,t)}{1-\alpha_t}\right) & \mathbf{x}_t = \mathbf{m}, \end{cases} \tag{8}$$

which implied that at each reverse step, we have a probability of $(\alpha_s - \alpha_t)/(1 - \alpha_t)$ to recover $x_0$, and once being recovered, $x_t$ stays fixed for the remainder of the process. Extensive works (Kingma et al., 2021; Shi et al., 2024) have shown that increasing discretization resolution can help approximate tighter evidence lower bound (ELBO). Therefore, we resort to optimizing the likelihood bound $\mathcal{L}_{\mathrm{cat}}$ under continuous time limit:

$$\mathcal{L}_{\mathrm{cat}}(\theta, k) = \mathbb{E}_q \int_{t=0}^{t=1} \frac{\alpha_t'}{1-\alpha_t} \mathbb{1}_{\{\mathbf{x}_t=\mathbf{m}\}} \log\langle \boldsymbol{\mu}_\theta^{\mathrm{cat}}(\mathbf{x}_t, t), \mathbf{x}_0^{\mathrm{cat}}\rangle dt, \tag{9}$$

where $\alpha_t'$ is the first order derivative of $\alpha_t$.

## 2.3 TRAINING WITH ADAPTIVELY LEARNABLE NOISE SCHEDULES

Tabular data is inherently highly heterogeneous of mixed numerical and categorical data types, and mixed feature distributions within each data type. Therefore, unlike pixels that share a similar distribution across three RGB channels and word tokens that share the exact same vocabulary space, each column (feature) of the table has its own specific marginal distributions, which requires the model to amortize its capacity adaptively across different features. We propose to adaptively learn a more fine-grained noise schedule for each feature respectively. To balance the trade-

---

**Algorithm 1** Training

1: **repeat**
2:     Sample $\mathbf{x}_0 \sim p_0(\mathbf{x})$
3:     Sample $t \sim U(0, 1)$
4:     Sample $\boldsymbol{\epsilon}_{\mathrm{num}} \sim \mathcal{N}(0, \mathbf{I}_{M\,\mathrm{num}})$
5:     $\mathbf{x}_t^{\mathrm{num}} = \mathbf{x}_0^{\mathrm{num}} + \boldsymbol{\sigma}^{\mathrm{num}}(t)\boldsymbol{\epsilon}_{\mathrm{num}}$
6:     $\boldsymbol{\alpha}_t = \exp(-\boldsymbol{\sigma}^{\mathrm{cat}}(t))$
7:     Sample $\mathbf{x}_t^{\mathrm{cat}} \sim q(\mathbf{x}_t|\mathbf{x}_0, \boldsymbol{\alpha}_t)$     Eq. (6)
8:     $\mathbf{x}_t = [\mathbf{x}_t^{\mathrm{num}}, \mathbf{x}_t^{\mathrm{cat}}]$
9:     Take gradient descent step on $\nabla_{\theta,\rho,k}\mathcal{L}_{\mathrm{TABDIFF}}$
10: **until** converged

---

off between the learnable noise schedule's flexibility and robustness, we design two function families: the power mean numerical schedule and the log-linear categorical schedule.

**Power-mean schedule for numerical features**. For the numerical noise schedule $\boldsymbol{\sigma}^{\mathrm{num}}(t)$ in Eq. (3), we define $\boldsymbol{\sigma}^{\mathrm{num}}(t) = [\sigma_{\rho_i}^{\mathrm{num}}(t)]$, with $\rho_i$ being a learnable parameter for individual numerical features. For $\forall i \in \{1, \cdots, M_{\mathrm{num}}\}$, we have $\sigma_{\rho_i}^{\mathrm{num}}(t)$ as:

$$\sigma_{\rho_i}^{\mathrm{num}}(t) = \left( \sigma_{\min}^{\frac{1}{\rho_i}} + t(\sigma_{\max}^{\frac{1}{\rho_i}} - \sigma_{\min}^{\frac{1}{\rho_i}}) \right)^{\rho_i}. \tag{10}$$

**Log-linear schedule for categorical features**. Similarly, for the categorical noise schedule $\boldsymbol{\sigma}^{\mathrm{cat}}(t)$ in Section 2.2, we define $\boldsymbol{\sigma}^{\mathrm{cat}}(t) = [\sigma_{k_j}^{\mathrm{cat}}(t)]$, with $k_i$ being a learnable parameter for individual categorical features. For $\forall j \in \{1, \cdots, M_{\mathrm{cat}}\}$, we have $\sigma_{k_j}^{\mathrm{cat}}(t)$ as:

$$\alpha_{k_j}^{\mathrm{cat}}(t) = 1 - t^{kj} \tag{11}$$

In practice, we fix the same initial and final noise levels across all numerical features so that $\sigma_i^{\mathrm{num}}(0) = \sigma_{\min}$ and $\sigma_i^{\mathrm{num}}(1) = \sigma_{\max}$ for $\forall i \in \{1, \cdots, M_{\mathrm{num}}\}$. We similarly bound the initial and final noise levels for the categorical features, as detailed in Appendix B.1. This enables us to constrain the freedom of schedules and thus stabilize the training.

**Joint objective function**. We update $M_{\mathrm{num}} + M_{\mathrm{cat}}$ parameters $\rho_1, \cdots, \rho_{M_{\mathrm{num}}}$ and $k_1, \cdots, k_{M_{\mathrm{cat}}}$ via backpropagation without the need of modifying the loss function. Consolidating $\mathcal{L}_{\mathrm{num}}$ and $\mathcal{L}_{\mathrm{cat}}$, we have the total loss $\mathcal{L}$ with two weight terms $\lambda_{\mathrm{num}}$ and $\lambda_{\mathrm{cat}}$ as:

$$\mathcal{L}_{\mathrm{TABDIFF}}(\theta, \rho, k) = \lambda_{\mathrm{num}} \mathcal{L}_{\mathrm{num}}(\theta, \rho) + \lambda_{\mathrm{cat}} \mathcal{L}_{\mathrm{cat}}(\theta, k)$$

$$= \mathbb{E}_{t \sim U(0,1)} \mathbb{E}_{(\mathbf{x}_t, \mathbf{x}_0) \sim q(\mathbf{x}_t, \mathbf{x}_0)} \left( \lambda_{\mathrm{num}} \|\boldsymbol{\mu}_\theta^{\mathrm{num}}(\mathbf{x}_t, t) - \boldsymbol{\epsilon}\|_2^2 + \frac{\lambda_{\mathrm{cat}} \alpha_t'}{1 - \alpha_t} \mathbb{1}_{\{\mathbf{x}_t = \mathbf{m}\}} \log \langle \boldsymbol{\mu}_\theta^{\mathrm{cat}}(\mathbf{x}_t, t), \mathbf{x}_0^{\mathrm{cat}} \rangle \right). \tag{12}$$

With the forward process defined in Eq. (3) and Eq. (6), we present the detailed training procedure in Algorithm 1. Here, we sample a continuous time step $t$ from a uniform distribution $U(0, 1)$ and then perturb numerical and categorical features with their respective noise schedules based on this same time index. Then, we input the concatenated $\mathbf{x}_t^{\mathrm{num}}$ and $\mathbf{x}_t^{\mathrm{cat}}$ into the model and take gradient on the joint loss function defined in Eq. (12).

## 2.4 SAMPLING WITH BACKWARD STOCHASTIC SAMPLER

One notable property of the joint sampling process is that the intermediate decoded categorical feature will not be updated anymore during sampling (see Eq. (8)). However, as tabular data are highly structured with complicated inter-column correlations, we expect the model to correct the error during sampling. To this end, we introduce a novel stochastic sampler by restarting the backward process with an additional forward process at each denoising step. Related work on continuous diffusions Karras et al. (2022); Xu et al. (2023) has shown that incorporating such stochasticity can yield better generation quality. We extend such intuition to both numerical and categorical features in tabular generation. At each sampling step $t$, we

---

**Algorithm 2** Sampling

1: Sample $\mathbf{x}_T^{\mathrm{num}} \sim \mathcal{N}(0, \mathbf{I}_{M\,\mathrm{num}})$, $\mathbf{x}_T^{\mathrm{cat}} = \boldsymbol{m}$
2: **for** $t = T$ to 1 **do**
3: $\quad t^+ \leftarrow t + \gamma_t t, \gamma_t = 1/T$
$\quad \triangleright$ Numerical forward perturbation:
4: $\quad$ Sample $\boldsymbol{\epsilon}^{\mathrm{num}} \sim \mathcal{N}(0, \mathbf{I}_{M\,\mathrm{num}})$
5: $\quad \mathbf{x}_{t^+}^{\mathrm{num}} \leftarrow \mathbf{x}_t^{\mathrm{num}} + \sqrt{\boldsymbol{\sigma}^{\mathrm{num}}(t^+)^2 - \boldsymbol{\sigma}^{\mathrm{num}}(t)^2} \boldsymbol{\epsilon}^{\mathrm{num}}$
$\quad \triangleright$ Categorical forward perturbation:
6: $\quad$ Sample $\mathbf{x}_{t^+}^{\mathrm{cat}} \sim q\left(\mathbf{x}_{t^+}^{\mathrm{cat}} | \mathbf{x}_t^{\mathrm{cat}}, 1 - \boldsymbol{\alpha}_{t^+}/\boldsymbol{\alpha}_t\right)$ $\quad$ Eq. (6)
$\quad \triangleright$ Concatenate:
7: $\quad \mathbf{x}_{t^+} = [\mathbf{x}_{t^+}^{\mathrm{num}}, \mathbf{x}_{t^+}^{\mathrm{cat}}]$
$\quad \triangleright$ Numerical backward ODE:
8: $\quad d\mathbf{x}^{\mathrm{num}} = (\mathbf{x}_{t^+}^{\mathrm{num}} - \mu_\theta^{\mathrm{num}}(\mathbf{x}_{t^+}, t^+))/\boldsymbol{\sigma}^{\mathrm{num}}(t^+)$
9: $\quad \mathbf{x}_{t-1}^{\mathrm{num}} \leftarrow \mathbf{x}_{t^+}^{\mathrm{num}} + (\boldsymbol{\sigma}^{\mathrm{num}}(t-1) - \boldsymbol{\sigma}^{\mathrm{num}}(t^+)) d\mathbf{x}^{\mathrm{num}}$
$\quad \triangleright$ Categorical backward sampling:
10: $\quad$ Sample $\mathbf{x}_{t-1}^{\mathrm{cat}} \sim p_\theta(\mathbf{x}_{t-1}^{\mathrm{cat}} | \mathbf{x}_{t^+}^{\mathrm{cat}}, \mu_\theta^{\mathrm{cat}}(\mathbf{x}_{t^+}, t^+))$ $\quad$ Eq. (8)
11: **end for**
12: **return** $\mathbf{x}_0^{\mathrm{num}}, \mathbf{x}_0^{\mathrm{cat}}$

---

first add a small time increment to the current time step $t$ to $t^+ = t + \gamma_t t$ according to a factor $\gamma_t$, and then perform the intermediate forward sampling between $t^+$ and $t$ by joint diffusion process Equations (3) and (6). From the increased-noise sample $\mathbf{x}_{t^+}$, we solve the ODE backward for $\mathbf{x}^{\mathrm{num}}$ and $\mathbf{x}^{\mathrm{cat}}$ from $t^+$ to $t - 1$, respectively, with a single update. This framework enables self-correction by randomly perturbing decoded features in the forward step. We summarize the sampling framework in Algorithm 2, and provide the ablation study for the stochastic sampler in Section 4.4. We also provide an illustrative example of the sampling process in Appendix C.

## 2.5 CLASSIFIER-FREE GUIDANCE CONDITIONAL GENERATION

TABDIFF can also be extended as a conditional generative model, which is important in many tasks such as missing value imputation. Let $\mathbf{y} = \{[\mathbf{y}^{\text{num}}, \mathbf{y}^{\text{cat}}]\}$ be the collection of provided properties in tabular data, containing both categorical and numerical features, and let $\mathbf{x}$ denote the missing interest features in this section. Imputation means we want to predict $\mathbf{x} = \{[\mathbf{x}^{\text{num}}, \mathbf{x}^{\text{cat}}]\}$ conditioned on $\mathbf{y}$. TABDIFF can be freely extended to conditional generation by only conducting denoising sampling for $\mathbf{x}_t$, while keeping other given features $\mathbf{y}_t$ fixed as $\mathbf{y}$.

Previous works on diffusion models (Dhariwal & Nichol, 2021) show that conditional generation quality can be further improved with a guidance classifier/regressor $p(\mathbf{y} \mid \mathbf{x})$. However, training the guidance classifier becomes challenging when $\mathbf{x}$ is a high-dimensional discrete object, and existing methods typically handle this by relaxing $\mathbf{x}$ as continuous (Vignac et al., 2023). Inspired by the classifier-free guidance (CFG) framework (Ho & Salimans, 2022) developed for continuous diffusion, we propose a unified CFG framework that eliminates the need for a classifier and handles mixed-type $\mathbf{x}$ and $\mathbf{y}$ effectively. The guided conditional sample distribution is given by $\tilde{p}_\theta(\mathbf{x}_t|\mathbf{y}) \propto p_\theta(\mathbf{x}_t|\mathbf{y})p_\theta(\mathbf{y}|\mathbf{x}_t)^\omega$, where $\omega > 0$ controls strength of the guidance. Applying Bayes' Rule, we get

$$\tilde{p}_\theta(\mathbf{x}_t|\mathbf{y}) \propto p_\theta(\mathbf{x}_t|\mathbf{y})p_\theta(\mathbf{y}|\mathbf{x}_t)^\omega = p_\theta(\mathbf{x}_t|\mathbf{y})\left(\frac{p_\theta(\mathbf{x}_t|\mathbf{y})p(\mathbf{y})}{p_\theta(\mathbf{x}_t)}\right)^\omega = \frac{p_\theta(\mathbf{x}_t|\mathbf{y})^{\omega+1}}{p_\theta(\mathbf{x}_t)^\omega}p(\mathbf{y})^\omega. \quad (13)$$

We drop $p(\mathbf{y})$ for it does no depend on $\theta$. Taking the logarithm of the probabilities, we obtain,

$$\log \tilde{p}_\theta(\mathbf{x}_t|\mathbf{y}) = (1+\omega)\log p_\theta(\mathbf{x}_t|\mathbf{y}) - \omega \log p_\theta(\mathbf{x}_t), \quad (14)$$

which implies the following changes in the sampling steps. For the numerical features, $\boldsymbol{\mu}_\theta^{\text{num}}(\mathbf{x}_t, t)$ is replaced by the interpolation of the conditional and unconditional estimates (Ho & Salimans, 2022):

$$\tilde{\boldsymbol{\mu}}_\theta^{\text{num}}(\mathbf{x}_t, \mathbf{y}, t) = (1+\omega)\boldsymbol{\mu}_\theta^{\text{num}}(\mathbf{x}_t, \mathbf{y}, t) - \omega\boldsymbol{\mu}_\theta^{\text{num}}(\mathbf{x}_t, t). \quad (15)$$

And for the categorical features, we instead predict $x_0$ with $\tilde{p}_\theta(\mathbf{x}_s^{\text{cat}}|\mathbf{x}_t, \mathbf{y})$, satisfying

$$\log \tilde{p}_\theta(\mathbf{x}_s^{\text{cat}}|\mathbf{x}_t, \mathbf{y}) = (1+\omega)\log p_\theta(\mathbf{x}_s^{\text{cat}}|\mathbf{x}_t, \mathbf{y}) - \omega \log p_\theta(\mathbf{x}_s^{\text{cat}}|\mathbf{x}_t). \quad (16)$$

Under the missing value imputation task, our target columns is $\mathbf{x}$, and the remaining columns constitute $\mathbf{y}$. Implementing CFG becomes very lightweight, as the guided probability utilizes the original unconditional model trained over all table columns as the conditional model and requires only an additional small model for the unconditional probabilities over the missing columns. We provide empirical results for CFG sampling in Section 4.3 and implementation details in Appendix B.2.

## 3 RELATED WORK

Recent studies have developed different generative models for tabular data, including VAE-based methods, TVAE (Xu et al., 2019) and GOGGLE (Liu et al., 2023), and GAN (Generative Adversarial Networks)-based methods, CTGAN (Xu et al., 2019) and TabelGAN (Park et al., 2018). These methods usually lack sufficient model expressivity for complicated data distribution. Recently, diffusion models have shown powerful generative ability for diverse data types and thus have been adopted by many tabular generation methods. Kotelnikov et al. (2023); Lee et al. (2023) designed separate discrete-time diffusion processes (Austin et al., 2021) for numerical and categorical features separately. However, they built their diffusion processes on discrete time steps, which have been proven to yield a looser ELBO estimation and thus lead to sub-optimal generation quality (Song et al., 2021; Kingma et al., 2021). To tackle such a problem caused by limited discretization of diffusion processes and push it to a continuous time framework, Zheng & Charoenphakdee (2022); Zhang et al. (2024) transform features into a latent continuous space via various encoding techniques, since advanced diffusion models are mainly designed for continuous random variables with Gaussian perturbation and thus cannot directly handle tabular data. However, it has shown that these solutions either are trapped with sub-optimal performance due to encoding overhead or cannot capture complex co-occurrence patterns of different modalities because of the indirect modeling and low model capacity. Concurrent work Mueller et al. (2024) also proposed feature-wise diffusion schedules, but the model still relies on encoding to continuous latent space with Gaussian diffusion framework. In summary, none of existing methods have explored the powerful mixed-type diffusion framework in the continuous-time limit and explicitly tackle the feature-wise heterogeneity issue in the mixed-type diffusion process.

# 4  EXPERIMENTS

We evaluate TABDIFF by comparing it to various generative models across multiple datasets and metrics, ranging from data fidelity and privacy to downstream task performance. Furthermore, we conduct ablation studies to investigate the effectiveness of each component of TABDIFF, e.g., the learnable noise schedules.

## 4.1  EXPERIMENTAL SETUPS

**Datasets**. We conduct experiments on seven real-world tabular datasets – Adult, Default, Shoppers, Magic, Faults, Beijing, News, and Diabetes – each containing both numerical and categorical attributes. In addition, each dataset has an inherent machine-learning task, either classification or regression. Detailed profiles of the datasets are presented in Appendix A.1.

**Baselines**. We compare the proposed TABDIFF with eight popular synthetic tabular data generation methods that are categorized into four groups: 1) GAN-based method: CTGAN (Xu et al., 2019); 2) VAE-based methods: TVAE (Xu et al., 2019) and GOGGLE (Liu et al., 2023); 3) Autoregressive Language Model: GReaT (Borisov et al., 2023); 4) Diffusion-based methods: STaSy (Kim et al., 2023), CoDi (Lee et al., 2023), TabDDPM (Kotelnikov et al., 2023) and TabSyn (Zhang et al., 2024).

**Evaluation Methods**. Following the previous methods (Zhang et al., 2024), we evaluate the quality of the synthetic data using eight distinct metrics categorized into three groups – 1) **Fidelity**: Shape, Trend, $\alpha$-Precision, $\beta$-Recall, and Detection assess how well the synthetic data can faithfully recover the ground-truth data distribution; 2) **Downstream tasks**: Machine learning efficiency and missing value imputation reveal the models' potential to power downstream tasks; 3) **Privacy**: The Distance to Closest Records (DCR) score evaluates the level of privacy protection by measuring how closely the synthetic data resembles the training data. We provide a detailed introduction of all these metrics in Appendix A.2.

**Implementation Details**. All reported experiment results are the average of 20 random sampled synthetic data generated by the best-validated models. Additional implementation details, such as the hardware/software information as well as hyperparameter settings, are in Appendix D.

## 4.2  DATA FIDELITY AND PRIVACY

**Shape and Trend**. We first evaluate the fidelity of synthetic data using the Shape and Trend metrics. Shape measures the synthetic data's ability to capture each single column's marginal density, while Trend assesses its capacity to replicate the correlation between different columns in the real data.

The detailed results for Shape and Trend metrics, measured across each dataset separately, are presented in Tables 1 and 2, respectively. On the Shape metric, TABDIFF outperforms all baselines on five out of seven datasets and surpasses the current state-of-the-art method TABSYN by an average of 13.3%. This demonstrates TABDIFF's superior performance in maintaining the marginal distribution of individual attributes across various datasets. Regarding the Trend metric, TABDIFF consistently outperforms all baselines and surpasses TABSYN by 22.6%. This significant improvement suggests that TABDIFF is substantially better at capturing column-column relationships than previous methods. Notably, TABDIFF maintains strong performance in Diabetes, a larger, more categorical-heavy dataset, surpassing the most competitive baseline by over 35% on both Shape and Trend. This exceptional performance thus demonstrates our model's capacity to model datasets with higher dimensionality and discrete features.

**Additional Fidelity Metrics**. We further evaluate the fidelity metrics across $\alpha$-precision, $\beta$-recall, and CS2T scores. On average, TABDIFF outperforms other methods on all these three metrics. We present the results for these three additional fidelity metrics in Appendix E.1.

**Data Privacy**. The ability to protect privacy is another important factor when evaluating synthetic data since we wish the synthetic data to be uniformly sampled from the data distribution manifold rather than being copied (or slightly modified) from each individual real data example. In this section, we use the Distance to Closest Records (DCR) score metric (Zhang et al., 2024), which measures the probability that a synthetic example's nearest neighbor is from a holdout v.s. the training set.

Table 1: Performance comparison on the error rates (%) of **Shape**.

| Method | Adult | Default | Shoppers | Magic | Beijing | News | Diabetes | Average |
|---|---|---|---|---|---|---|---|---|
| CTGAN | $16.84_{\pm 0.03}$ | $16.83_{\pm 0.04}$ | $21.15_{\pm 0.10}$ | $9.81_{\pm 0.08}$ | $21.39_{\pm 0.05}$ | $16.09_{\pm 0.02}$ | $9.82_{\pm 0.08}$ | 15.99 |
| TVAE | $14.22_{\pm 0.08}$ | $10.17_{\pm 0.05}$ | $24.51_{\pm 0.06}$ | $8.25_{\pm 0.06}$ | $19.16_{\pm 0.06}$ | $16.62_{\pm 0.03}$ | $18.86_{\pm 0.13}$ | 15.97 |
| GOGGLE | 16.97 | 17.02 | 22.33 | 1.90 | 16.93 | 25.32 | 24.92 | 17.91 |
| GReaT | $12.12_{\pm 0.04}$ | $19.94_{\pm 0.06}$ | $14.51_{\pm 0.12}$ | $16.16_{\pm 0.09}$ | $8.25_{\pm 0.12}$ | OOM | OOM | 14.20 |
| STaSy | $11.29_{\pm 0.06}$ | $5.77_{\pm 0.06}$ | $9.37_{\pm 0.09}$ | $6.29_{\pm 0.13}$ | $6.71_{\pm 0.03}$ | $6.89_{\pm 0.03}$ | OOM | 7.72 |
| CoDi | $21.38_{\pm 0.06}$ | $15.77_{\pm 0.07}$ | $31.84_{\pm 0.05}$ | $11.56_{\pm 0.26}$ | $16.94_{\pm 0.02}$ | $32.27_{\pm 0.04}$ | $21.13_{\pm 0.25}$ | 21.55 |
| TabDDPM | $1.75_{\pm 0.03}$ | $1.57_{\pm 0.08}$ | $2.72_{\pm 0.13}$ | $1.01_{\pm 0.09}$ | $1.30_{\pm 0.03}$ | $78.75_{\pm 0.01}$ | $31.44_{\pm 0.05}$ | 16.93 |
| TABSYN [1] | $0.81_{\pm 0.05}$ | $1.01_{\pm 0.08}$ | $1.44_{\pm 0.07}$ | $1.03_{\pm 0.14}$ | $1.26_{\pm 0.05}$ | $2.06_{\pm 0.04}$ | $1.85_{\pm 0.02}$ | 1.35 |
| TABDIFF | $0.63_{\pm 0.05}$ | $1.24_{\pm 0.07}$ | $1.28_{\pm 0.09}$ | $0.78_{\pm 0.08}$ | $1.03_{\pm 0.05}$ | $2.35_{\pm 0.03}$ | $0.89_{\pm 0.23}$ | 1.17 |
| Improv. | 22.2% ↓ | 0.0% ↓ | 11.11% ↓ | 14.29% ↓ | 18.25% ↓ | 0% ↓ | 46.39% ↓ | 13.3% ↓ |

[1] TABSYN's performance is obtained via our reproduction. The results of other baselines except on Diabetes, are taken from Zhang et al. (2024). The OOM entries are explained in Appendix D.

Table 2: Performance comparison on the error rates (%) of **Trend**.

| Method | Adult | Default | Shoppers | Magic | Beijing | News | Diabetes | Average |
|---|---|---|---|---|---|---|---|---|
| CTGAN | $20.23_{\pm 1.20}$ | $26.95_{\pm 0.93}$ | $13.08_{\pm 0.16}$ | $7.00_{\pm 0.19}$ | $22.95_{\pm 0.08}$ | $5.37_{\pm 0.05}$ | $18.95_{\pm 0.34}$ | 16.36 |
| TVAE | $14.15_{\pm 0.88}$ | $19.50_{\pm 0.95}$ | $18.67_{\pm 0.38}$ | $5.82_{\pm 0.49}$ | $18.01_{\pm 0.08}$ | $6.17_{\pm 0.09}$ | $32.74_{\pm 0.26}$ | 16.44 |
| GOGGLE | 45.29 | 21.94 | 23.90 | 9.47 | 45.94 | 23.19 | 27.56 | 28.18 |
| GReaT | $17.59_{\pm 0.22}$ | $70.02_{\pm 0.12}$ | $45.16_{\pm 0.18}$ | $10.23_{\pm 0.40}$ | $59.60_{\pm 0.55}$ | OOM | OOM | 44.24 |
| STaSy | $14.51_{\pm 0.25}$ | $5.96_{\pm 0.26}$ | $8.49_{\pm 0.15}$ | $6.61_{\pm 0.53}$ | $8.00_{\pm 0.10}$ | $3.07_{\pm 0.04}$ | OOM | 7.77 |
| CoDi | $22.49_{\pm 0.08}$ | $68.41_{\pm 0.05}$ | $17.78_{\pm 0.11}$ | $6.53_{\pm 0.25}$ | $7.07_{\pm 0.15}$ | $11.10_{\pm 0.01}$ | $29.21_{\pm 0.12}$ | 23.21 |
| TabDDPM | $3.01_{\pm 0.25}$ | $4.89_{\pm 0.10}$ | $6.61_{\pm 0.16}$ | $1.70_{\pm 0.22}$ | $2.71_{\pm 0.09}$ | $13.16_{\pm 0.11}$ | $51.54_{\pm 0.05}$ | 11.95 |
| TABSYN | $1.93_{\pm 0.07}$ | $2.81_{\pm 0.48}$ | $2.13_{\pm 0.10}$ | $0.88_{\pm 0.18}$ | $3.13_{\pm 0.34}$ | $1.52_{\pm 0.03}$ | $3.90_{\pm 0.04}$ | 2.33 |
| TABDIFF | $1.49_{\pm 0.16}$ | $2.55_{\pm 0.75}$ | $1.74_{\pm 0.08}$ | $0.76_{\pm 0.12}$ | $2.59_{\pm 0.15}$ | $1.28_{\pm 0.04}$ | $2.20_{\pm 0.16}$ | 1.80 |
| Improve. | 22.8% ↓ | 9.3% ↓ | 18.3% ↓ | 13.6% ↓ | 4.4% ↓ | 15.8% ↓ | 37.3% ↓ | 22.6% ↓ |

Due to space limits, the explanations for the additional fidelity metrics and data privacy metrics, along with the corresponding experiments, are deferred to Appendices A.2 and E.

### 4.3 PERFORMANCE ON DOWNSTREAM TASKS

**Machine Learning Efficiency.** A key advantage of high-quality synthetic data is its ability to serve as an anonymized proxy for real datasets and power effective learning on downstream tasks such as classification and regression. We measure the synthetic table's capacity to support downstream task learning via Machine Learning Efficiency (MLE). Following established protocols (Kim et al., 2023; Lee et al., 2023; Xu et al., 2019), we first split the real dataset into training and test sets, then train the given generative model on the real training set. Subsequently, we sample a synthetic dataset of equal size to the real training set from the models and use it to train an XGBoost Classifier or XGBoost Regressor (Chen & Guestrin, 2016). Finally, we evaluate these machine learning models against the real test set to calculate the AUC score and RMSE for classification and regression tasks, respectively.

According to the MLE results presented in Table 3, TABDIFF consistently achieves the best or second-best performance across all datasets, with the highest average performance outperforming the most competitive baseline TABSYN by 15.0%. This demonstrates our method's competitive capacity to capture and replicate key features of the real data that are most relevant to learning downstream machine learning tasks. However, while TABDIFF shows strong performance on MLE, we observe that methods with varying performance on data fidelity metrics might have very close MLE scores. This suggests that the MLE score evaluated under the current setting may not be a reliable indicator of data quality. Therefore, we complement MLE with additional quality metrics in Appendix E, which better highlights the superior performance of TABDIFF.

**Missing Value Imputation.** We further evaluate TABDIFF's conditional generation capacity through the Missing Value Imputation task. Following the approach in Zhang et al. (2024), we treat the inherent classification/regression task of each dataset as an imputation task. Specifically, for each table, we train generative models on the training set to generate the target column while conditioning on the remaining columns. The imputation performance is measured by the model's accuracy in recovering the target column of the test set. Implementing classifier-free guidance (CFG) for this

Table 3: Evaluation of **MLE** (Machine Learning Efficiency): AUC and RMSE are used for classification and regression tasks, respectively.

| Methods | Adult | Default | Shoppers | Magic | Beijing | News[1] | Diabetes | Average Gap |
|---|---|---|---|---|---|---|---|---|
| | AUC ↑ | AUC ↑ | AUC ↑ | AUC ↑ | RMSE ↓ | RMSE ↓ | AUC ↑ | % |
| Real | $.927_{\pm.000}$ | $.770_{\pm.005}$ | $.926_{\pm.001}$ | $.946_{\pm.001}$ | $.423_{\pm.003}$ | $.842_{\pm.002}$ | $.704_{\pm.002}$ | 0.0 |
| CTGAN | $.886_{\pm.002}$ | $.696_{\pm.005}$ | $.875_{\pm.009}$ | $.855_{\pm.006}$ | $.902_{\pm.019}$ | $.880_{\pm.016}$ | $.569_{\pm.004}$ | 23.7 |
| TVAE | $.878_{\pm.004}$ | $.724_{\pm.005}$ | $.871_{\pm.006}$ | $.887_{\pm.003}$ | $.770_{\pm.011}$ | $1.01_{\pm.016}$ | $.594_{\pm.009}$ | 20.2 |
| GOGGLE | $.778_{\pm.012}$ | $.584_{\pm.005}$ | $.658_{\pm.052}$ | $.654_{\pm.024}$ | $1.09_{\pm.025}$ | $.877_{\pm.002}$ | $.475_{\pm.008}$ | 42.1 |
| GReaT | $.913_{\pm.003}$ | $.755_{\pm.006}$ | $.902_{\pm.005}$ | $.888_{\pm.008}$ | $.653_{\pm.013}$ | OOM | OOM | 13.3 |
| STaSy | $.906_{\pm.001}$ | $.752_{\pm.006}$ | $.914_{\pm.005}$ | $.934_{\pm.003}$ | $.656_{\pm.014}$ | $.871_{\pm.002}$ | OOM | 10.9 |
| CoDi | $.871_{\pm.006}$ | $.525_{\pm.006}$ | $.865_{\pm.006}$ | $.932_{\pm.003}$ | $.818_{\pm.021}$ | $1.21_{\pm.005}$ | $.505_{\pm.004}$ | 30.2 |
| TabDDPM | $.907_{\pm.001}$ | $.758_{\pm.004}$ | $.918_{\pm.005}$ | $.935_{\pm.003}$ | $.592_{\pm.011}$ | $4.86_{\pm3.04}$ | $.521_{\pm.008}$ | 11.95 |
| TABSYN | $.909_{\pm.001}$ | $\mathbf{.763}_{\pm\mathbf{.002}}$ | $.914_{\pm.004}$ | $\mathbf{.937}_{\pm\mathbf{.002}}$ | $.580_{\pm.009}$ | $\mathbf{.862}_{\pm\mathbf{.024}}$ | $.684_{\pm.002}$ | 6.78 |
| TABDIFF | $\mathbf{.912}_{\pm\mathbf{.002}}$ | $\mathbf{.763}_{\pm\mathbf{.005}}$ | $\mathbf{.921}_{\pm\mathbf{.004}}$ | $.936_{\pm.003}$ | $\mathbf{.555}_{\pm\mathbf{.013}}$ | $.866_{\pm.021}$ | $\mathbf{.689}_{\pm\mathbf{.016}}$ | **5.76** |

Table 4: Performance of TABDIFF in the Missing Value Imputation task. We draw a direct comparison to the generative approach employed by TABSYN, with the performance of XGBoost classifiers/regressors included as a reference.

| Methods | Adult | Default | Shoppers | Magic | Beijing | News | Diabetes | Avg. Improv. |
|---|---|---|---|---|---|---|---|---|
| | AUC ↑ | AUC ↑ | AUC ↑ | AUC ↑ | RMSE ↓ | RMSE ↓ | AUC ↑ | % |
| Predicted by XGBoost | 92.7 | 77.0 | 92.6 | 94.6 | 0.423 | 0.842 | 70.4 | 0.0 |
| Impute with TABSYN | 93.1 | 86.7 | **96.5** | 91.3 | **0.386** | 0.818 | 66.6 | 4.99 |
| Impute with TABDIFF + CFG $(\omega = 0.0)$ | 92.5 | 91.6 | 95.7 | 92.5 | 0.424 | 0.828 | 66.0 | 3.76 |
| Impute with TABDIFF + CFG $(\omega = 0.6)$ | **93.2** | **91.7** | 96.4 | **93.0** | 0.414 | **0.815** | **66.9** | **5.60** |

task is straightforward. We approximate the conditional model using the unconditioned TABDIFF trained on all columns from the previous unconditional generation tasks. For the unconditional model, we train TABDIFF on the target column with a significantly smaller denoising network. Detailed implementation is provided in Appendix D, and results are presented in Table 4.

As demonstrated, TABDIFF achieves higher imputation accuracy than TABSYN on five out of seven datasets, with an average improvement of $5.60\%$ over the non-generative XGBoost classifier. This indicates TABDIFF's superior capacity for conditional tabular data generation. Moreover, we empirically demonstrate the efficacy of our CFG framework by showing that the model consistently performs better with $\omega = 0.6$ compared to $\omega = 0.0$ (which is equivalent to TABDIFF without CFG).

## 4.4 ABLATION STUDIES

**Stochastic Sampler.** We conduct ablation studies to assess the effectiveness of the stochastic sampler, discussed in Section 2.4. The results are presented in Table 5. We use 'Det.' and 'Sto.' as abbreviations for deterministic and stochastic samplers. The deterministic sampler refers to the conventional diffusion backward process described in Song et al. (2021); Karras et al. (2022), consisting of a series of deterministic ODE steps. According to Table 5, under both fixed and learnable noise schedules, TABDIFF with the stochastic sampler consistently outperforms the deterministic sampler on the fidelity metrics Shape and Trend, regardless of whether learnable noise schedules are enabled. These confirm the efficacy of additional stochasticity in reducing decoding errors during backward diffusion sampling.

**Adaptively Learnable Noise Schedule.** Next, we perform an ablation study to evaluate the effectiveness of our adaptively learnable noise schedules. We compare the model with learnable schedules against the model with non-learnable noise schedules, where the noise parameters for numerical features are fixed to $\rho_i \equiv 7, \forall i$ in Eq. (10) and, for numerical features, fixed to $k_j \equiv 1, \forall j$ in Eq. (11). We refer to these models as 'Learn.' and 'Fix.', respectively. According to the results in Table 5, the learnable noise schedules substantially improve performance, particularly in Trend and regardless of whether the stochastic sampler is enabled. Furthermore, we closely examine the training process of both models on the Adult dataset by plotting their changes of training loss in Figure 2. According to the figure, the learnable schedules (orange curves) significantly reduce both numerical and categorical losses in Eq. (12).

| Method | Shape | Trend |
|---|---|---|
| TABSYN | 1.35 | 2.33 |
| TABDIFF-Fix.+Det. | 1.39 | 2.29 |
| TABDIFF-Fix.+Sto. | 1.20 | 1.93 |
| TABDIFF-Learn.+Det. | 1.24 | 1.92 |
| TABDIFF-Learn.+Sto. | **1.17** | **1.80** |

Table 5: Ablation Studies on the stochastic sampler and learnable noise schedules.

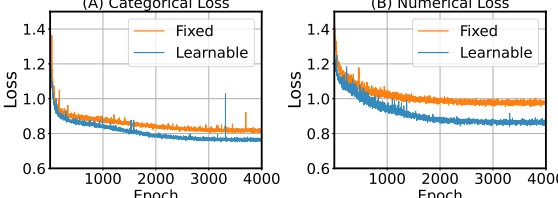

Figure 2: The adaptively learnable noise schedules reduce training loss.

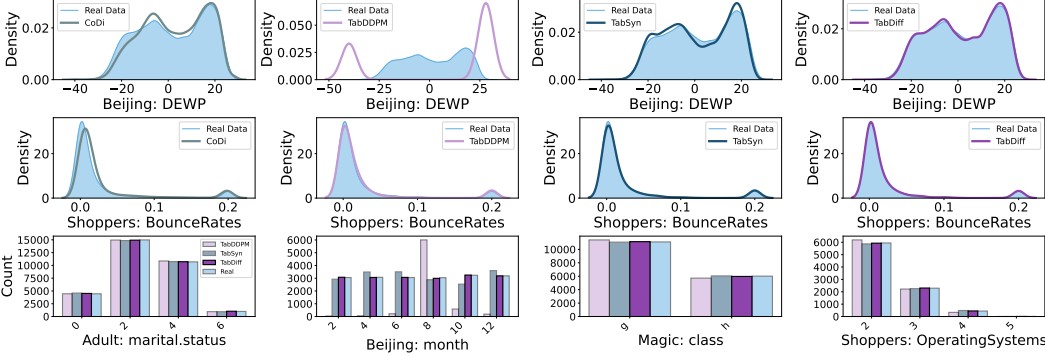

Figure 3: Visualization of the marginal densities of the generated data in comparison to the real data. Top and Middle: individual numerical column; Bottom: individual categorical column.

## 4.5 VISUALIZATIONS OF SYNTHETIC DATA

We present a comprehensive set of visualizations to compare single-column marginal distributions and pairwise correlations across four models—CoDi, TabDDPM, TABSYN, and our TABDIFF—and four distinct datasets: Adult, Beijing, Magic, and Shoppers. In Figure 3, we provide 1-dimensional kernel density estimation (KDE) curves for a chosen numerical feature, alongside histograms for a chosen categorical feature. According to the figures, the density of TABDIFF's samples matches most closely with that of the real data, highlighting TABDIFF's ability to capture the original distribution patterns. Furthermore, in Figure 4, we include correlation heatmaps that show the correlation error rate for each pair of columns. These pictures consistently demonstrate that the TABDIFF archives the closest match to real correlation scores, highlighting its superior ability to capture the column-wise correlation of the real data.

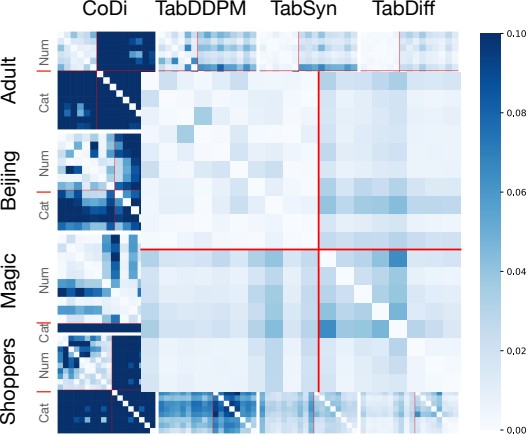

Figure 4: Pair-wise correlation heatmaps. Values represent the error rate (the lighter, the better).

## 5 CONCLUSION

In this paper, we have introduced TABDIFF, a mixed-type diffusion framework for generating high-quality synthetic data. TABDIFF combines a hybrid diffusion process to handle numerical and categorical features in their native formats. To address the disparate distributions of features and their interrelationships, we further introduced several key innovations, including learnable column-wise noise schedules and the stochastic sampler. We conducted extensive experiments using a diverse set of datasets and metrics, comprehensively comparing TABDIFF with existing approaches. The results demonstrate TABDIFF's superior capacity in learning the original data distribution and generating faithful and diverse synthetic data to power downstream tasks.

ACKNOWLEDGMENT

We gratefully acknowledge the support of NSF under Nos. OAC-1835598 (CINES), CCF-1918940 (Expeditions), DMS-2327709 (IHBEM), IIS-2403318 (III); Stanford Data Applications Initiative, Wu Tsai Neurosciences Institute, Stanford Institute for Human-Centered AI, Chan Zuckerberg Initiative, Amazon, Genentech, GSK, Hitachi, SAP, and UCB. We also gratefully acknowledge the support of ARO (W911NF-21-1-0125), ONR (N00014-23-1-2159), and Chan Zuckerberg Biohub. Minkai Xu thanks the generous support of Sequoia Capital Stanford Graduate Fellowship.

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

# A DETAILED EXPERIMENT SETUPS

## A.1 DATASETS

We use seven tabular datasets from UCI Machine Learning Repository[1]: Adult, Default, Shoppers, Magic, Beijing, News, and Diabetes, where each tabular dataset is associated with a machine-learning task. Classification: Adult, Default, Magic, Shoppers, and Diabetes. Regression: Beijing and News. The statistics of the datasets are presented in Table 6.

Table 6: Statistics of datasets. # Num stands for the number of numerical columns, and # Cat stands for the number of categorical columns. # Max Cat stands for the number of categories of the categorical column with the most categories.

| Dataset | # Rows | # Num | # Cat | # Max Cat | # Train | # Validation | # Test | Task |
|---------|--------|-------|-------|-----------|---------|--------------|--------|------|
| **Adult** | $48,842$ | 6 | 9 | 42 | $28,943$ | $3,618$ | $16,281$ | Classification |
| **Default** | $30,000$ | 14 | 11 | 11 | $24,000$ | $3,000$ | $3,000$ | Classification |
| **Shoppers** | $12,330$ | 10 | 8 | 20 | $9,864$ | $1,233$ | $1,233$ | Classification |
| **Magic** | $19,019$ | 10 | 1 | 2 | $15,215$ | $1,902$ | $1,902$ | Classification |
| **Beijing** | $43,824$ | 7 | 5 | 31 | $35,058$ | $4,383$ | $4,383$ | Regression |
| **News** | $39,644$ | 46 | 2 | 7 | $31,714$ | $3,965$ | $3,965$ | Regression |
| **Diabetes** | $101,766$ | 9 | 27 | 716 | $61,059$ | $2,0353$ | $20,354$ | Classification |

## A.2 METRICS

### A.2.1 SHAPE AND TREND

Shape and Trend are proposed by SDMetrics[2]. They are used to measure the column-wise density estimation performance and pair-wise column correlation estimation performance, respectively. Shape uses Kolmogorov-Sirnov Test (KST) for numerical columns and the Total Variation Distance (TVD) for categorical columns to quantify column-wise density estimation. Trend uses Pearson correlation for numerical columns and contingency similarity for categorical columns to quantify pair-wise correlation.

**Shape**. *Kolmogorov-Sirnov Test (KST)*: Given two (continuous) distributions $p_r(x)$ and $p_s(x)$ ($r$ denotes real and $s$ denotes synthetic), KST quantifies the distance between the two distributions using the upper bound of the discrepancy between two corresponding Cumulative Distribution Functions (CDFs):

$$\text{KST} = \sup_x |F_r(x) - F_s(x)|, \tag{17}$$

where $F_r(x)$ and $F_s(x)$ are the CDFs of $p_r(x)$ and $p_s(x)$, respectively:

$$F(x) = \int_{-\infty}^{x} p(x)\mathrm{d}x. \tag{18}$$

*Total Variation Distance*: TVD computes the frequency of each category value and expresses it as a probability. Then, the TVD score is the average difference between the probabilities of the categories:

$$\text{TVD} = \frac{1}{2} \sum_{\omega \in \Omega} |R(\omega) - S(\omega)|, \tag{19}$$

where $\omega$ describes all possible categories in a column $\Omega$. $R(\cdot)$ and $S(\cdot)$ denotes the real and synthetic frequencies of these categories.

**Trend**. *Pearson Correlation Coefficient*: The Pearson correlation coefficient measures whether two continuous distributions are linearly correlated and is computed as:

$$\rho_{x,y} = \frac{\text{Cov}(x,y)}{\sigma_x \sigma_y}, \tag{20}$$

---

[1] https://archive.ics.uci.edu/datasets
[2] https://docs.sdv.dev/sdmetrics

where $x$ and $y$ are two continuous columns. Cov is the covariance, and $\sigma$ is the standard deviation.

Then, the performance of correlation estimation is measured by the average differences between the real data's correlations and the synthetic data's corrections:

$$\text{Pearson Score} = \frac{1}{2}\mathbb{E}_{x,y}|\rho^R(x,y) - \rho^S(x,y)|, \tag{21}$$

where $\rho^R(x,y)$ and $\rho^S(x,y))$ denotes the Pearson correlation coefficient between column $x$ and column $y$ of the real data and synthetic data, respectively. As $\rho \in [-1, 1]$, the average score is divided by 2 to ensure that it falls in the range of $[0, 1]$, then the smaller the score, the better the estimation.

*Contingency similarity*: For a pair of categorical columns $A$ and $B$, the contingency similarity score computes the difference between the contingency tables using the Total Variation Distance. The process is summarized by the formula below:

$$\text{Contingency Score} = \frac{1}{2}\sum_{\alpha \in A}\sum_{\beta \in B}|R_{\alpha,\beta} - S_{\alpha,\beta}|, \tag{22}$$

where $\alpha$ and $\beta$ describe all the possible categories in column $A$ and column $B$, respectively. $R_{\alpha,\beta}$ and $S_{\alpha,\beta}$ are the joint frequency of $\alpha$ and $\beta$ in the real data and synthetic data, respectively.

### A.2.2  $\alpha$-PRECISION AND $\beta$-RECALL

Following Liu et al. (2023) and Alaa et al. (2022), we adopt the $\alpha$-Precision and $\beta$-Recall proposed in Alaa et al. (2022), two sample-level metric quantifying how faithful the synthetic data is. In general, $\alpha$-Precision evaluates the fidelity of synthetic data – whether each synthetic example comes from the real-data distribution, $\beta$-Recall evaluates the coverage of the synthetic data, e.g., whether the synthetic data can cover the entire distribution of the real data (In other words, whether a real data sample is close to the synthetic data.)

### A.2.3  DETECTION

The detection measures the difficulty of detecting the synthetic data from the real data when they are mixed. We use the classifer-two-sample-test (C2ST) implemented by SDMetrics, where a logistic regression model plays the role of a detector.

### A.2.4  MACHINE LEARNING EFFICIENCY

In MLE, each dataset is first split into the real training and testing set. The generative models are learned on the real training set. After training, a synthetic set of equivalent size is sampled.

The performance of synthetic data on MLE tasks is evaluated based on the divergence of test scores using the real and synthetic training data. Therefore, we first train the machine learning model on the real training set, split into training and validation sets with a $8 : 1$ ratio. The classifier/regressor is trained on the training set, and the optimal hyperparameter setting is selected according to the performance on the validation set. After the optimal hyperparameter setting is obtained, the corresponding classifier/regressor is retrained on the training set and evaluated on the real testing set. The performance of synthetic data is obtained in the same way.

## B  METHOD DETAILS

### B.1  ADAPTIVELY LEARNABLE NOISE SCHEDULES

For numerical stability, we need to bound $\sigma_{\min}$ and $\sigma_{\max}$ within $(0, 1)$. As shown in Eq. (10), our formulation of the power-mean noise schedule boundaries the noise level in between $\sigma_{\min}$ and $\sigma_{\max}$. To make sure that the noise level for numerical features is also bounded, we linearly map $t$ to the interval $[\delta, 1 - \delta]$, thus recasting Eq. (11) into

$$\sigma_{k_j}^{\text{cat}}(t) = -\log\left(1 - \left((1 - \delta) \cdot t^{k_j} + \delta\right)\right). \tag{23}$$

### B.2   CLASSIFIER-FREE GUIDANCE

In this section, we elaborate on how to implement our classifier-free guided conditional generation.

**Simple way to compute** $\tilde{p}_\theta(\mathbf{x}_s^{\mathrm{cat}}|\mathbf{x}_t, \mathbf{y})$**.** We first show that, under our simple masked diffusion framework, the guided posterior probability for categorical columns, $\tilde{p}_\theta(\mathbf{x}_s^{\mathrm{cat}}|\mathbf{x}_t, \mathbf{y})$ can be simply computed by directly interpolating the model's raw estimates of $\mathbf{x}_0$, i.e., $\boldsymbol{\mu}_\theta^{\mathrm{cat}}(\mathbf{x}_t, \mathbf{y}, t)$ and $\boldsymbol{\mu}_\theta^{\mathrm{cat}}(\mathbf{x}_t, t)$.

If $\mathbf{x}_t$ is already unmasked (i.e., $\mathbf{x} = \mathbf{m}$), we remain at the current state as before. Otherwise, we compute the posterior according to Eq. (16). Note that all operations below are performed element-wise.

$$\log \tilde{p}_\theta(\mathbf{x}_s^{\mathrm{cat}}|\mathbf{x}_t, \mathbf{y}) = (1+\omega)\log p_\theta(\mathbf{x}_s^{\mathrm{cat}}|\mathbf{x}_t, \mathbf{y}) - \omega \log p_\theta(\mathbf{x}_s^{\mathrm{cat}}|\mathbf{x}_t).$$

$$\tilde{p}_\theta(\mathbf{x}_s^{\mathrm{cat}}|\mathbf{x}_t, \mathbf{y}) = \frac{p_\theta(\mathbf{x}_s^{\mathrm{cat}}|\mathbf{x}_t, \mathbf{y})^{\omega+1}}{p_\theta(\mathbf{x}_s^{\mathrm{cat}}|\mathbf{x}_t)^\omega}$$

$$= \frac{\left(\frac{(1-\alpha_s)\mathbf{m} + (\alpha_s - \alpha_t)\boldsymbol{\mu}_\theta^{\mathrm{cat}}(\mathbf{x}_t, \mathbf{y}, t)}{1-\alpha_t}\right)^{\omega+1}}{\left(\frac{(1-\alpha_s)\mathbf{m} + (\alpha_s - \alpha_t)\boldsymbol{\mu}_\theta^{\mathrm{cat}}(\mathbf{x}_t, t)}{1-\alpha_t}\right)^\omega}$$

$$= \frac{\left((1-\alpha_s)\mathbf{m} + (\alpha_s - \alpha_t)\boldsymbol{\mu}_\theta^{\mathrm{cat}}(\mathbf{x}_t, \mathbf{y}, t)\right)^{\omega+1}}{\left((1-\alpha_s)\mathbf{m} + (\alpha_s - \alpha_t)\boldsymbol{\mu}_\theta^{\mathrm{cat}}(\mathbf{x}_t, t)\right)^\omega} \frac{1}{1-\alpha_t}$$

Since $\boldsymbol{\mu}_\theta^{\mathrm{cat}}$ and $\mathbf{m}$ must have zero probability mass in each other's dimension that can have a positive mass, the exponent of summations into the summation of exponents:

$$= \frac{((1-\alpha_s)\mathbf{m})^{\omega+1} + ((\alpha_s - \alpha_t)\boldsymbol{\mu}_\theta^{\mathrm{cat}}(\mathbf{x}_t, \mathbf{y}, t))^{\omega+1}}{((1-\alpha_s)\mathbf{m})^\omega + ((\alpha_s - \alpha_t)\boldsymbol{\mu}_\theta^{\mathrm{cat}}(\mathbf{x}_t, t))^\omega} \frac{1}{1-\alpha_t}$$

By the same property, we can perform division for $\mathbf{m}$ and $\boldsymbol{\mu}_\theta^{\mathrm{cat}}$ separately:

$$= \left((1-\alpha_s)\mathbf{m} + (\alpha_s - \alpha_t)\frac{\boldsymbol{\mu}_\theta^{\mathrm{cat}}(\mathbf{x}_t, \mathbf{y}, t)^{\omega+1}}{\boldsymbol{\mu}_\theta^{\mathrm{cat}}(\mathbf{x}_t, t)^\omega}\right) \frac{1}{1-\alpha_t}$$

$$= \frac{(1-\alpha_s)\mathbf{m} + (\alpha_s - \alpha_t)\exp\big((1+\omega)\log\boldsymbol{\mu}_\theta^{\mathrm{cat}}(\mathbf{x}_t, \mathbf{y}, t) - \omega\log\boldsymbol{\mu}_\theta^{\mathrm{cat}}(\mathbf{x}_t, t)\big)}{1-\alpha_t}$$

Therefore, we can formulate $\tilde{p}_\theta(\mathbf{x}_s^{\mathrm{cat}}|\mathbf{x}_t, \mathbf{y})$ as

$$\tilde{p}_\theta(\mathbf{x}_s^{\mathrm{cat}}|\mathbf{x}_t, \mathbf{y}) = \begin{cases} \mathrm{cat}(\mathbf{x}_s^{\mathrm{cat}}; \mathbf{x}_t^{\mathrm{cat}}) & \mathbf{x}_t^{\mathrm{cat}} \neq \mathbf{m}, \\ \mathrm{cat}\left(\mathbf{x}_s^{\mathrm{cat}}; \frac{(1-\alpha_s)\mathbf{m} + (\alpha_s - \alpha_t)\tilde{\boldsymbol{\mu}}_\theta^{\mathrm{cat}}(\mathbf{x}_t, t)}{1-\alpha_t}\right) & \mathbf{x}_t = \mathbf{m}, \end{cases}$$

where $\tilde{\boldsymbol{\mu}}_\theta^{\mathrm{cat}}(\mathbf{x}_t, t)$ can be simply computed as the interpolation of $\boldsymbol{\mu}_\theta^{\mathrm{cat}}(\mathbf{x}_t, \mathbf{y}, t)$ and $\boldsymbol{\mu}_\theta^{\mathrm{cat}}(\mathbf{x}_t, t)$:

$$\log \tilde{\boldsymbol{\mu}}_\theta^{\mathrm{cat}}(\mathbf{x}_t, t) = (1+\omega)\log\boldsymbol{\mu}_\theta^{\mathrm{cat}}(\mathbf{x}_t, \mathbf{y}, t) - \omega\log\boldsymbol{\mu}_\theta^{\mathrm{cat}}(\mathbf{x}_t, t)$$

## C   DETAILED ILLUSTRATIONS OF TRAINING AND SAMPLING PROCESSES

**Training details.** Algorithm 1 outlines the training procedure for our hybrid diffusion model that jointly processes numerical and categorical variables. At each training iteration, we begin by sampling an initial data point $\mathbf{x}_0$ from the training data distribution $p(\mathbf{x_0})$ and a timestep $t$ uniformly from $[0, 1]$. Then, we perform the forward diffusion step. Each numerical feature is perturbed with a Gaussian noise whose intensity is determined by the $\boldsymbol{\sigma}^{\mathrm{num}}(t)$; each categorical feature flipped into the mask token $\mathbf{m}$ with probability $\boldsymbol{\alpha}_t$ (i.e., eq. (6)). The noised numerical and categorical components are concatenated together to form a noisy version $\mathbf{x}_t$ of the table row. Lastly, we compute the training objective $\mathcal{L}_{\mathrm{TABDIFF}}$ and perform gradient descent on the model parameters $\theta$, $\rho$, and $k$.

**Sampling details.** Here, we present a vivid visual example in Figure 5 to illustrate the backward sampling process described in Algorithm 2. Our example demonstrates generating a table with

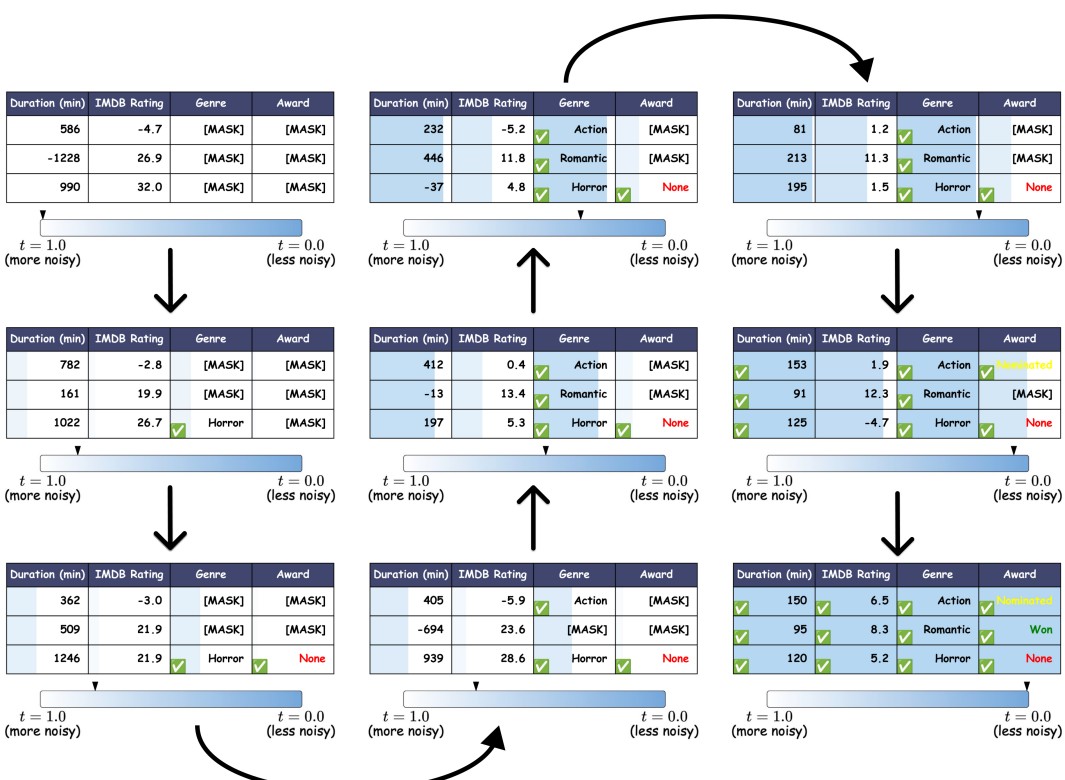

Figure 5: A vivid visualization of TABDIFF's generation process.

two numerical columns (movie duration and IMDB rating) and two categorical columns (genre and awards status). Each row represents an independent sample.

First, at $t = 1.0$ (the first frame), the numerical features are initialized with Gaussian noise, and all categorical components are masked. The algorithm then iterates backward through time steps from $t = 1.0$ to $0.0$.

At each timestep $t$, we first perform the forward stochastic perturbation step (the yellow section of Algorithm 2), the core of our stochastic sampler. All features are first perturbed forward to $t^+$, a slightly noisier state, following the same process as the forward step during training. While this step is not explicitly depicted in our visualization in Figure 5, it implies that, for instance, during the transition at the third frame, the unmasked "None" entry could be stochastically flipped back to the [MASK] state. This would then allow the model to re-predict the value, potentially yielding a different result (e.g "Won") than "None" in the subsequent frame.

After the stochastic perturbation, we perform the denoising/unmasking step (the blue section of Algorithm 2). For numerical features, we denoise to $\mathbf{x}_{t-1}^{\text{num}}$ by solving an ODE The update delta is determined by the normalized difference, $d\mathbf{x}^{\text{num}}$, between the current state and the model's prediction, scaled by the decrease in noise levels. For categorical variables, we perform the unmasking step. Intuitively, if the column is already unmasked, we stay in the current state. This is demonstrated in Figure 5, where the "None" entry persists once it has been flipped. If it is still masked, we flip the mask token to some valid value of that column with a certain probability ($\frac{\alpha_{t-1} - \alpha_t}{1 - \alpha_t}$) that increases as sampling proceeds. We choose which unmasked token to move to based on the model's predicted probability $\mu_{\theta}^{\text{cat}}(x_t, t)$ over all possible categories of the column.

# D  IMPLEMENTATION DETAILS

We perform our experiment on an Nvidia RTX A4000 GPU with 16G memory and implement TABDIFF with PyTorch.

**Data preprocessing.** The raw tabular datasets usually contain missing entries. Thus in the first phase of preprocessing, we make up these missing values in the same way as existing works (Kotelnikov et al., 2023; Zhang et al., 2024), with numerical missing values being replaced by the column average and categorical missing values being treated as a new category. Moreover, the diverse range of numerical features typically leads to more difficult and unstable training. To counter this, we then transform the numerical values with the QuantileTransformer[3], and recover the original values using its inverse during sampling.

**Data splits.** For datasets other than Diabetes, we follow the exact same split as Zhang et al. (2024). Each dataset is split into the "real" and "test" sets. For the unconditional generation task on which data fidelity and the imputation task, the models are trained on the "real" set and evaluated on the "real" set. For the MLE task, the "real" set is further split into a training and validation set, and the "test" set is used for testing. Finally, for the data privacy measure DCR, the original dataset is equally split into two halves, with one being treated as the training set and the other as the holdout set.

For Diabetes, we split it into train, validation, and test sets with a ratio of 0.6/0.2/0.2. For the MLE task. The training and test sets are regarded as the "real" and "test" sets for the unconditional generation and imputation tasks. For DCR, we apply an equal split.

**Architecture of the denoising network.** In our implementation, we project each column individually to a $d$ dimensional vector using a linear layer, ensuring that all columns are treated with the same importance. We set the embedding size $d$ as 4, matching the size used in Zhang et al. (2024). We then process these projected vectors with a two-layer transformer, appending positional encodings at the end. The transformed vectors are then concatenated and passed through a five-layer MLP conditioned on the time embedding. Finally, the output is obtained by sequentially applying another transformer followed by a projection layer that recovers the original dimensions. Our denoising network has a comparable number of parameters as those experimented in TabDDPM (Kotelnikov et al., 2023) and TABSYN (Zhang et al., 2024), as our shared MLP model accounts for most of the parameters.

**Hyperparameters Setting.** TABDIFF employs the same hyperparameter setting for all datasets. We train our models for 8000 epochs with the Adam optimizer. The training and sampling batch sizes are set to 4,096 and 10,000, respectively. Regarding the hyperparameters in TABDIFF, the values $\sigma_{\min}$ and $\sigma_{\max}$ are set to $0.002$ and $80.0$, referencing the optimal setting in Karras et al. (2022), and $\delta$ are set to $1e{-}3$. For the loss weightings, we fix $\lambda_{\mathrm{cat}}$ to 1.0 and linear decay $\lambda_{\mathrm{num}}$ from 1.0 to 0.0 as training proceeds.

During inference, we select the checkpoint with the lowest training loss. We observe that our model achieves the superior performance reported paper with as few as 50 discretization steps ($T = 50$).

**Details on OOMs in experiment result tables**:

1. GOOGLE set fixed random seed during sampling in the official codes, and we follow it for consistency.

2. GReaT cannot be applied on News for maximum length limit.

3. STaSy runs out of memory on Diabetes that has hight cardinality categorical columns

4. TabDDPM cannot produce meaningful content on the News and Diabetes datasets.

**Imputation.** As mentioned in Section 4.3, we obtain the unconditional model of the target column by training TABDIFF on it with a smaller denoising network. For this network, we keep the same architecture but reduce the number of MLP layers to one.

# E  DETAILED EXPERIMENTS RESULTS

In the following sections, we discuss the result on $\alpha$-precision, $\beta$-recall, detection score (C2ST), and DCR in detail.

---

[3]https://scikit-learn.org/stable/modules/generated/sklearn.preprocessing.QuantileTransformer.html

### E.1 ADDITIONAL FIDELITY METRICS

**$\alpha$-precision.** We first evaluate TABDIFF on $\alpha$-Precision score, a metric that measures the quality of synthetic data. Higher scores indicate the synthetic data is more faithful to the real. We present the results across all seven datasets in Table 7. TABDIFF achieves the best or second-best performance on all datasets. Specifically, TABDIFF ranks first with an average $\alpha$-Precision score of 98.22, surpassing all other baseline methods.

**$\beta$-recall.** Next, we compare TABDIFF to the baselines on the $\beta$-Recall scores, which evaluates the extent to which synthetic data covers the real data distribution. The results are presented in Table 8, with a higher score reflecting a more comprehensive coverage of the real data's feature space. TABDIFF consistently outperforms or matches the top-performing baselines, achieving the highest average $\beta$-Recall score of 49.40. This indicates that the generated data spans a broad range of the real distribution. Though some baseline methods attained higher scores on specific datasets, they fail to demonstrate competitive performance on $\alpha$-Precision, as models have to trade off fine-grained details in order to capture a broader range of features.

Overall, TABDIFF maintains a balance between broad data coverage and preserving fine-grained details. This balance highlights TABDIFF 's capability in generating synthetic data that faithfully captures both the breadth and depth of the original data distribution.

**Detection Score (C2ST).** Lastly, we assess the fidelity of synthetic data by using the C2ST test, which evaluates how difficult it is to distinguish the synthetic data from the real data. The results are shown in Table 9, where a higher score indicates better fidelity. TABDIFF achieves the best performance on five of seven datasets, outperforming the most competitive baseline model by $6.89\%$ on average. Notably, TABDIFF excels on Diabetes, which contains many numerous high-cardinality categorical features (as indicated by *# Max Cat* in Table 6), showcasing its ability to generate high-quality categorical data. These results, therefore, demonstrate TABDIFF's capacity to generate synthetic data that closely resembles the real data.

Table 7: Comparison of $\alpha$-Precision scores. **Bold Face** highlights the best score for each dataset. Higher scores reflect better performance.

| Methods | Adult | Default | Shoppers | Magic | Beijing | News | Diabetes | Average | Ranking |
|---|---|---|---|---|---|---|---|---|---|
| CTGAN | $77.74_{\pm 0.15}$ | $62.08_{\pm 0.08}$ | $76.97_{\pm 0.39}$ | $86.90_{\pm 0.22}$ | $96.27_{\pm 0.14}$ | $96.96_{\pm 0.17}$ | $79.89_{\pm 0.10}$ | 82.40 | 5 |
| TVAE | $98.17_{\pm 0.17}$ | $85.57_{\pm 0.34}$ | $58.19_{\pm 0.26}$ | $86.19_{\pm 0.48}$ | $97.20_{\pm 0.10}$ | $86.41_{\pm 0.17}$ | $19.24_{\pm 0.15}$ | 75.85 | 7 |
| GOGGLE | 50.68 | 68.89 | 86.95 | 90.88 | 88.81 | 86.41 | 23.09 | 70.81 | 9 |
| GReaT | $55.79_{\pm 0.03}$ | $85.90_{\pm 0.17}$ | $78.88_{\pm 0.13}$ | $85.46_{\pm 0.54}$ | $\mathbf{98.32_{\pm 0.22}}$ | OOM | OOM | 80.87 | 6 |
| STaSy | $82.87_{\pm 0.26}$ | $90.48_{\pm 0.11}$ | $89.65_{\pm 0.25}$ | $86.56_{\pm 0.19}$ | $89.16_{\pm 0.12}$ | $94.76_{\pm 0.33}$ | OOM | 88.91 | 3 |
| CoDi | $77.58_{\pm 0.45}$ | $82.38_{\pm 0.15}$ | $94.95_{\pm 0.35}$ | $85.01_{\pm 0.36}$ | $98.13_{\pm 0.38}$ | $87.15_{\pm 0.12}$ | $64.80_{\pm 0.53}$ | 84.29 | 4 |
| TabDDPM | $96.36_{\pm 0.20}$ | $97.59_{\pm 0.36}$ | $88.55_{\pm 0.68}$ | $98.59_{\pm 0.17}$ | $97.93_{\pm 0.30}$ | $0.00_{\pm 0.00}$ | $28.35_{\pm 0.11}$ | 72.48 | 8 |
| TABSYN | $\mathbf{99.39_{\pm 0.18}}$ | $\mathbf{98.65_{\pm 0.23}}$ | $98.36_{\pm 0.52}$ | $99.42_{\pm 0.28}$ | $97.51_{\pm 0.24}$ | $95.05_{\pm 0.30}$ | $\mathbf{96.61_{\pm 0.24}}$ | 97.86 | 2 |
| TABDIFF | $99.02_{\pm 0.20}$ | $98.49_{\pm 0.28}$ | $\mathbf{99.11_{\pm 0.34}}$ | $\mathbf{99.47_{\pm 0.21}}$ | $98.06_{\pm 0.24}$ | $\mathbf{97.36_{\pm 0.17}}$ | $95.69_{\pm 0.19}$ | **98.22** | 1 |

Table 8: Comparison of $\beta$-Recall scores. **Bold Face** highlights the best score for each dataset. Higher scores reflects better results.

| Methods | Adult | Default | Shoppers | Magic | Beijing | News | Diabetes | Average | Ranking |
|---|---|---|---|---|---|---|---|---|---|
| CTGAN | $30.80_{\pm 0.20}$ | $18.22_{\pm 0.17}$ | $31.80_{\pm 0.350}$ | $11.75_{\pm 0.20}$ | $34.80_{\pm 0.10}$ | $24.97_{\pm 0.29}$ | $9.42_{\pm 0.26}$ | 23.11 | 8 |
| TVAE | $38.87_{\pm 0.31}$ | $23.13_{\pm 0.11}$ | $19.78_{\pm 0.10}$ | $32.44_{\pm 0.35}$ | $28.45_{\pm 0.08}$ | $29.66_{\pm 0.21}$ | $4.92_{\pm 0.13}$ | 25.32 | 7 |
| GOGGLE | 8.80 | 14.38 | 9.79 | 9.88 | 19.87 | 2.03 | 3.74 | 9.78 | 9 |
| GReaT | $49.12_{\pm 0.18}$ | $42.04_{\pm 0.19}$ | $44.90_{\pm 0.17}$ | $34.91_{\pm 0.28}$ | $43.34_{\pm 0.31}$ | OOM | OOM | 43.34 | 3 |
| STaSy | $29.21_{\pm 0.34}$ | $39.31_{\pm 0.39}$ | $37.24_{\pm 0.45}$ | $53.97_{\pm 0.57}$ | $54.79_{\pm 0.18}$ | $39.42_{\pm 0.32}$ | OOM | 42.32 | 4 |
| CoDi | $9.20_{\pm 0.15}$ | $19.94_{\pm 0.22}$ | $20.82_{\pm 0.23}$ | $50.56_{\pm 0.31}$ | $52.19_{\pm 0.12}$ | $34.40_{\pm 0.31}$ | $2.70_{\pm 0.06}$ | 27.12 | 6 |
| TabDDPM | $47.05_{\pm 0.25}$ | $47.83_{\pm 0.35}$ | $47.79_{\pm 0.25}$ | $\mathbf{48.46_{\pm 0.42}}$ | $56.92_{\pm 0.13}$ | $0.00_{\pm 0.00}$ | $0.03_{\pm 0.01}$ | 35.44 | 5 |
| TABSYN | $47.92_{\pm 0.23}$ | $46.45_{\pm 0.35}$ | $49.10_{\pm 0.60}$ | $48.03_{\pm 0.50}$ | $59.15_{\pm 0.22}$ | $\mathbf{43.01_{\pm 0.28}}$ | $33.72_{\pm 0.16}$ | 46.77 | 2 |
| TABDIFF | $\mathbf{51.64_{\pm 0.20}}$ | $\mathbf{51.09_{\pm 0.25}}$ | $\mathbf{49.75_{\pm 0.64}}$ | $48.01_{\pm 0.31}$ | $\mathbf{59.63_{\pm 0.23}}$ | $42.10_{\pm 0.32}$ | $\mathbf{41.74_{\pm 0.17}}$ | **49.40** | 1 |

### E.2 DATA PRIVACY.

Table 10 shows the DCR scores across all datasets. The DCR score represents the probability that a generated data sample is more similar to the training set than to the test set, with a score closer to 50% being ideal, as it indicates a balance between the similarity to training and test distributions.

Table 9: Detection score (C2ST) using logistic regression classifier. Higher scores reflect superior performance.

| Method | Adult | Default | Shoppers | Magic | Beijing | News | Diabetes | Average |
|--------|-------|---------|----------|-------|---------|------|----------|---------|
| CTGAN | 0.5949 | 0.4875 | 0.7488 | 0.6728 | 0.7531 | 0.6947 | 0.5593 | 0.6444 |
| TVAE | 0.6315 | 0.6547 | 0.2962 | 0.7706 | 0.8659 | 0.4076 | 0.0487 | 0.5250 |
| GOGGLE | 0.1114 | 0.5163 | 0.1418 | 0.9526 | 0.4779 | 0.0745 | 0.0912 | 0.3380 |
| GReaT | 0.5376 | 0.4710 | 0.4285 | 0.4326 | 0.6893 | OOM | OOM | 0.5118 |
| STaSy | 0.4054 | 0.6814 | 0.5482 | 0.6939 | 0.7922 | 0.5287 | OOM | 0.6083 |
| CoDi | 0.2077 | 0.4595 | 0.2784 | 0.7206 | 0.7177 | 0.0201 | 0.0008 | 0.3435 |
| TabDDPM | 0.9755 | 0.9712 | 0.8349 | **0.9998** | 0.9513 | 0.0002 | 0.1980 | 0.7044 |
| TABSYN | 0.9910 | **0.9826** | 0.9662 | 0.9960 | 0.9528 | 0.9255 | 0.5953 | 0.9156 |
| TABDIFF | **0.9950** | 0.9774 | **0.9843** | 0.9989 | **0.9781** | **0.9308** | **0.9865** | **0.9787** |
| Improv. | 0.40% ↓ | 0.0% ↓ | 1.87% ↓ | 0.0% ↓ | 2.66% ↓ | 0.57% ↓ | 65.71% ↓ | 6.89% ↓ |

Table 10: DCR score, which represents the probability that a generated data sample is more similar to the training set than to the test set. A score closer to $50\%$ is more preferable. **Bold Face** highlights the best score for each dataset.

| Method | Adult | Default | Shoppers | Beijing | News | Diabetes |
|--------|-------|---------|----------|---------|------|----------|
| STaSy | $50.33\%_{\pm 0.19}$ | $50.23\%_{\pm 0.09}$ | $51.53\%_{\pm 0.16}$ | $50.59\%_{\pm 0.29}$ | $50.59\%_{\pm 0.14}$ | OOM |
| CoDi | $49.92\%_{\pm 0.18}$ | $51.82\%_{\pm 0.26}$ | $51.06\%_{\pm 0.18}$ | $50.87\%_{\pm 0.11}$ | $50.79\%_{\pm 0.23}$ | $51.12\%_{\pm 0.19}$ |
| TabDDPM | $51.14\%_{\pm 0.18}$ | $52.15\%_{\pm 0.20}$ | $63.23\%_{\pm 0.25}$ | $80.11\%_{\pm 2.68}$ | $79.31\%_{\pm 0.29}$ | $37.76\%_{\pm 0.23}$ |
| TABSYN | $50.94\%_{\pm 0.17}$ | $51.20\%_{\pm 0.18}$ | $52.90\%_{\pm 0.22}$ | $50.37\%_{\pm 0.13}$ | $50.85\%_{\pm 0.33}$ | $50.62\%_{\pm 0.28}$ |
| TABDIFF | $50.10\%_{\pm 0.32}$ | $51.11\%_{\pm 0.36}$ | $50.24\%_{\pm 0.62}$ | $50.50\%_{\pm 0.36}$ | $51.04\%_{\pm .32}$ | $50.43\%_{\pm 0.18}$ |

Across the datasets, TABDIFF consistently achieves DCR scores near 50%, highlighting its ability to generalize well while maintaining fidelity to the original data distribution.

# F STUDY OF MODEL EFFICIENCY AND ROBUSTNESS

In this section, we present a thorough analysis of TabDDPM's efficiency and robustness. For efficiency, we compare the training and sampling speed of TABDIFF against other competitive baseline methods (TabDDPM, TABSYN) using **four different metrics**. For robustness, we first explore the tradeoff between discretization steps (i.e., efficiency) and sample quality in diffusion-based models. We then dig into detailed error rates Shape and Trend to see whether models are biased towards learning some particular columns of datasets. Lastly, we discussed the robustness issues we found with the competitive baseline. We use the representative Adult dataset, which contains a balanced number of numerical and categorical columns, throughout the experiment. Our results show that, among the competitive methods, TABDIFF is not only the most effective but also the most robust and highly efficient. Below, we present a detailed analysis.

## F.1 EFFICIENCY

**Training Time.** We first measure the training time of each method. The epoch lengths are set to the default configuration of each method, and the validation frequencies are set to the same value of once every 200 epochs. Our measurements are presented in the first column of the Table 11, with the entry of TABSYN being split into – VAE training time + Diffusion training time.

We see that all three methods take a comparable time to complete one training run, with TABSYN being 10% faster than TABDIFF and TABDIFF being 20% faster than TabDDPM. The current gap between TABDIFF and TABSYN is likely due to TABDIFF 's slightly deeper network architecture compared to TABSYN. We believe that the architecture of TABDIFF can be further optimized to improve efficiency, and we leave this as future work.

Nevertheless, it is also important to consider model robustness when assessing efficiency. As highlighted in Appendix F.2, the training process for TABSYN is notably unstable due to the difficulty

| Method | Train.T (min) | Sample.T (sec) | NFE |
|--------|--------------|----------------|-----|
| TabDDPM | 112 | $125.1_{\pm 0.01}$ | 1 |
| TABSYN | $45 + 40 = 85$ | $8.8_{\pm 0.005}$ | 2 |
| TABDIFF | 94 | $15.2_{\pm 0.007}$ | 2 |

Table 11: Model Efficiency.

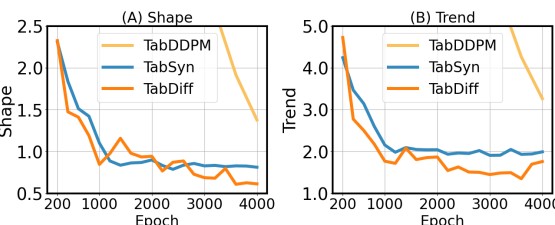

Figure 6: Model Convergence speed.

in training VAEs, often requiring you to retrain many times in order to produce a model capable of generating samples comparable in quality to TABDIFF. On the other hand, TabDDPM fails to scale to more complicated datasets as shown by its poor performance on News and Diabetes. Thus, when taking into account training robustness, TABDIFF is the most robust and efficient model among all competitive methods.

**Training Convergence.** Next, we assess training convergence by evaluating the quality of samples generated from intermediate checkpoints during the training process. Figure 6 plots our result. The curves show that TABDIFF converges faster than the other methods, as TABDIFF produces more high-quality samples when shown to the datasets for the same number of times (i.e., at a same epoch).

**Number of Function Evaluation (NFE).** For sampling, we first theoretically analyze the number of network calls involved in a single diffusion step (i.e. the denoising step from $x_t$ to $x_{t-1}$)) for each method. The numbers are shown in the third column of Table 11. TABDIFF and TABSYN involve an extra network call because TABDIFF employs the second-order correction trick introduced in Karras et al. (2022).

**Sampling Time.** We empirically measure the time to generate the same number of samples as the test set (32561 samples for Adult). The numbers are presented in the second column of Table 11. According to them, TABDIFF and TABSYN samples $\sim 10\times$ faster than TabDDPM. This result is expected, as both TABDIFF and TABSYN, by default, use 20 times fewer sampling steps than TabDDPM, while making twice as many network calls per step compared to TabDDPM. TABDIFF 's slightly longer sampling time is also attributed to the evaluation of its deeper network. We believe this can be optimized in future work.

## F.2 ROBUSTNESS

**Controling quality-efficiency tradeoff through discretization steps.** One advantage of continuous-time diffusion models (which currently include TABDIFF and TABSYN) is the ability to sample with arbitrary discretization steps, allowing them to flexibly tradeoff sample quality with sample efficiency. We conduct an experiment that compares how TABDIFF and TABSYN perform when sampled with different discretization steps. The results and their visualizations are presented in Table 12 and Figure 7.

Our results show that TABDIFF consistently achieves higher sample quality across all levels of discretization steps. Notably, when the number of steps is reduced to just 5 (requiring only one second for sampling), TABSYN fails to generate meaningful content, as indicated by an error rate approaching 50%. In contrast, TABDIFF continues to produce valid and coherent content even under these highly constrained conditions.

**Evaluting column-wise learning bias** To address what it means by "optimal allocation of model capacity across heterogeneous features," we analyze how generation quality varies between different features. The Shape and Trend errors presented in Table 5 are averaged over all columns and columns pars. Now, using the representative Adult dataset, we visualize the errors at each column and column pair in Figures 8 and 9, along with the normalized standard deviations in Table 13 to quantitatively measure the variations of errors. All results show that TABDIFF with learnable schedules not only achieves lower average error but also exhibits more consistent errors across columns. This balance indicates that learnable schedules help the model balance its capacity on different dimensions of the data, improving the model's ability to handle heterogeneous distributions.

|  | **TABSYN** | | **TABDIFF** | |
|---|---|---|---|---|
| **Steps** | **Shape** | **Trend** | **Shape** | **Trend** |
| 5 | 34.09 | 49.30 | 12.51 | 22.15 |
| 10 | 1.99 | 3.92 | 1.55 | 3.36 |
| 25 | 0.84 | 1.96 | 0.62 | 1.50 |
| 50 | 0.81 | 1.95 | 0.63 | 1.49 |
| 100 | 0.82 | 1.94 | 0.64 | 1.53 |

Table 12: Ablate sample steps.

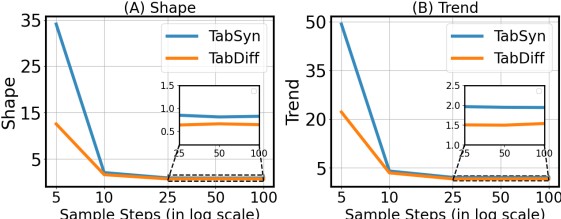

Figure 7: Visualize the sample step ablation

| **Method** | **Shape** | **Trend** | **Shape Std.** | **Trend Std.** |
|---|---|---|---|---|
| TABSYN | 0.81 | 1.93 | 1.01 | 0.88 |
| TABDIFF-Fixed | 0.74 | 1.73 | 0.42 | 0.75 |
| TABDIFF-Learn. | **0.63** | **1.49** | **0.29** | **0.64** |

Table 13: Evaluating column-wise learning bias.

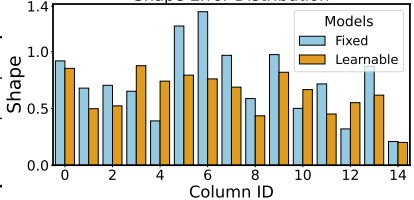

Figure 8: TABDIFF with learnable schedules has a more balanced Shape performance.

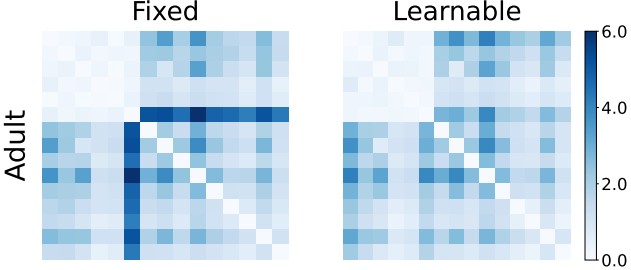

Figure 9: TABDIFF with learnable schedules has a more balanced Trend performance

**Robustness issues of baselines.** We identify several robustness issues with the competitive baseline methods. Specifically, TabDDPM struggles to scale to larger datasets, and TABSYN 's performance is highly dependent on the training quality of VAEs, which can vary significantly across different runs.

TabDDPM: As shown by the results in Tables 1 and 2, TabDDPM achieves poor performance on larger datasets like News and Diabetes. This is because it failed to generate meaningful samples, as we can see in Figure 11 that the numerical columns of TabDDPM's Diabetes samples all collapsed to the minimal and maximal values of the domains. After examining the training logs, we discovered that this poor generation performance might be due to the explosion of training loss, as shown in Figure 10.

TABSYN: TABSYN is another competitive tabular generation model whose performance, to our best knowledge, is closest to TABDIFF. However, this method has a limitation: as mentioned in Zhang et al. (2024), the quality of the VAE has a significant impact on TABSYN 's performance, as it's the only component that recovers the original data shape. When reproducing TABSYN's result, we observed that across different training runs, the sample quality varies significantly. For poorly performing runs, we attempted to retrain the diffusion model and even increased the number of training epochs, but these efforts did not improve the results. This confirms that the issue lies with the VAE.

To further illustrate this, we present the results of Shape and Trend that are averaged across 10 random training runs in Table 14 (Note: in the paper, we follow the convention of previous works and reported results based on 20 different runs of the same checkpoint, and we put the result of the best reproduction run for TABSYN). These additional results demonstrate that TABDIFF achieves

| Method | Adult | Default | Beijing |
|---|---|---|---|
| TABSYN (Shape) | $1.31_{\pm0.64}$ | $\mathbf{1.17}_{\pm0.21}$ | $2.69_{\pm1.44}$ |
| TABSYN (Trend) | $2.73_{\pm0.98}$ | $5.05_{\pm2.22}$ | $5.05_{\pm1.88}$ |
| TABDIFF (Shape) | $\mathbf{0.65}_{\pm0.08}$ | $1.19_{\pm0.12}$ | $\mathbf{1.07}_{\pm0.6}$ |
| TABDIFF (Trend) | $\mathbf{1.47}_{\pm0.18}$ | $\mathbf{2.46}_{\pm0.62}$ | $\mathbf{2.61.}_{\pm0.20}$ |

Table 14: Models' consistency across different training runs.

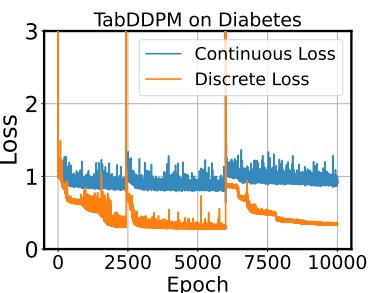

Figure 10: TabDDPM failed to converge on Diabetes

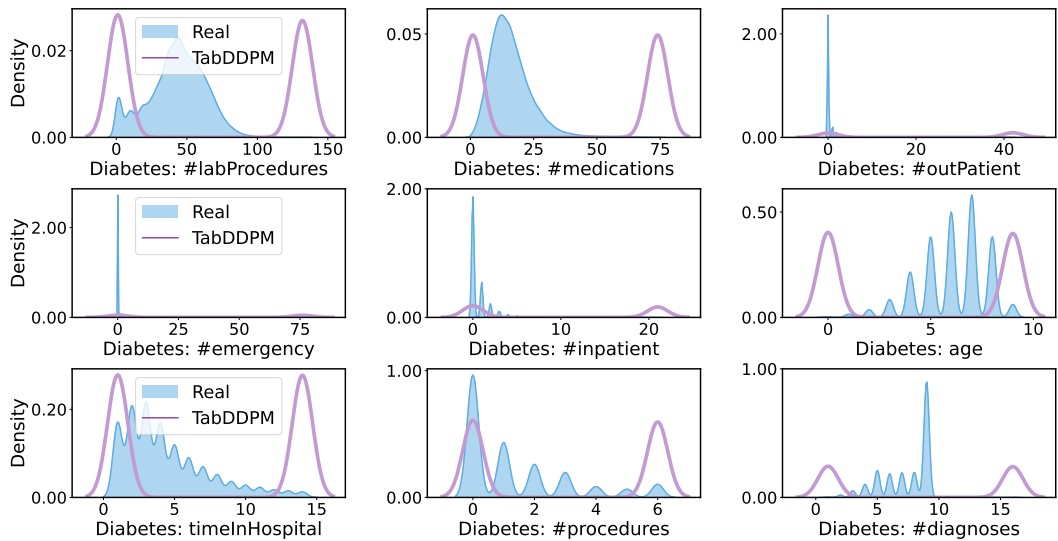

Figure 11: TabDDPM failed to produce meaningful samples on Diabetes

significantly more consistent performance across different training runs, as shown by the smaller average and standard deviation.

