# OpenReview forum: "TabDiff: a Mixed-type Diffusion Model for Tabular Data Generation"
_ICLR.cc/2025/Conference — ICLR 2025 Poster_

### Official Review · Reviewer_Buu2 · 2024-10-31

**Soundness:** 3
**Presentation:** 3
**Contribution:** 3
**Rating:** 6
**Confidence:** 3

**Summary:**

This paper introduces TABDIFF, a joint diffusion framework which aims to handle data with both discrete and continuous data types. The authors validated the effectiveness of this framework through comprehensive experiments.

**Strengths:**

1.The method introduces a new approach by jointly learning the diffusion processes for discrete and continuous features, which is not explored in the previous work.

2. The adaptively learnable noise schedules look novel.

3.The paper provides extensive experimental evaluations across different benchmark datasets, thoroughly validating the method’s performance.

**Weaknesses:**

In the title of this paper, it is claimed that this paper addresses multi-modal distributions. However, the paper does not clearly specify what multi-modal distribution it handles. Does it only refer to distributions with both discrete and continuous variables?

**Questions:**

NA

---

> ### Author Response · Authors · 2024-11-20
> **Author Response Part 1**
>
> Thank you for your constructive feedback and insightful questions! Below, we provide detailed responses to each of your questions. Additionally, please refer to our revised paper, with updates colored in purple.
>
> >**[W1] Modeling feature dependencies**
>
> Thank you for raising the point regarding the conditional independence assumption made when estimating the posterior $q(x_s|x_t,x_0)$.
>
> **Factorization does not limit learning inter-feature dependencies.** We first clarify that this assumption allows for easy optimization while still allowing the model to capture complex dependencies between different features. In the diffusion model, it’s a common practice to factorize the posterior estimation across all data dimensions. In practice, we not only factorize between numerical and categorical features but factorize across all individual features, in the same fashion as image diffusion models factorize across all pixels in the image grids. The detailed formulation is shown below:
>
> $$
> q(x\_s \mid x\_t, x\_0) = \prod\_{i=1}^{M\_\text{num}} q((x\_s^\text{num})\_i, \mid x\_t, x\_0)
> \cdot \prod\_{j=1}^{M\_\text{cat}} q((x\_s^\text{cat})\_j, \mid x\_t, x\_0)
> $$
>
> This factorization does not imply that the numerical and categorical features are independent in the reverse generative process nor dependencies are ignored. This is because each conditional distribution $q(x_s^{\text{num}} \mid x_t, x_0)$  and $q(x_s^{\text{cat}} \mid x_t, x_0)$ is conditioned on the entire $x_t$ and $x_0$, which contains information from both numerical and categorical features. This allows the model to learn and capture complex dependencies between different features.
>
> The efficacy of this approach in capturing cross-feature dependencies has been proven across many applications of diffusion models. Examples include modeling local spatial dependencies in image generation [1,2] and modeling the dependency between atom coordinates and types in molecular generation [3]. In the context of tabular data generation, this assumption is also effective in capturing dependencies between features of different types, as evidenced by its adoption in methods like TabDDPM, TabSyn, and ours.
>
> **How TabDiff’s design tricks further improve dependency modeling.**  Beyond this standard practice, TabDiff introduces feature-wise adaptive noise schedules that further enhance the model’s ability to capture dependencies between features. These learnable schedules give the model the flexibility to denoise different features at varying speeds. Specifically, a more positive $\rho$ and $k$ corresponds to quicker denoising. During training, the model learns different denoising rates for different features feature, which can be regarded as an order of generation. Features with larger $\rho$ and $k$ are recovered earlier in the denoising process, allowing subsequent features to be generated and conditioned on them. This mechanism enables the model to better capture inter–feature dependencies. In Table 2, TabDiff outperforms all other baselines on inter-column correlation metrics (i.e., Trend); in Table 5, TabDiff with learnable schedules achieves higher scores on Trend.
>
> [1] Ho et al., 2020. “Denoising Diffusion Probabilistic Models.”
>
> [2] Karras et al., 2022. “Elucidating the Design Space of Diffusion-Based Generative Models.”
>
> [3] Hoogeboom et al., 2022. “Elucidating the Design Space of Diffusion-Based Generative Models.”

---

> > ### Author Response · Authors · 2024-11-20
> > **Author Response Part 2**
> >
> > >**[W2] TabDiff vs. TabSyn on data fidelity**
> >
> > **Our results show TabDiff is better in terms of data fidelity.** We respectfully disagree with the assertion that TabDiff underperforms TabSyn in data fidelity. In fact, our results show that TabDiff consistently outperforms TabSyn on average across all data fidelity metrics.
> >
> > Specifically, for the most direct metrics of data fidelity — column-wise density and inter-column correlation (i.e. Shape and Trend — TabDiff achieves substantial improvements over TabSyn, with average improvements of 13.3% and 22.5%,  respectively. Notably, TabDiff achieves higher correlation scores across all datasets.
> >
> > These improvements are clearly illustrated in Figures 3 and 4. In Figure 3,  the density curves of TabDiff’s samples align more closely with those of the real samples, indicating superior fidelity in column-wise distributions. Similarly, in Figure 4, the heatmaps for TabDiff’s samples are significantly lighter than Tabsyn’s, implying that the intercolumn correlations of TabDiff’s samples are closer to that of the real ones.
> >
> > TabDiff also surpasses TabSyn on additional fidelity metrics that we discussed in detail in Appendix D.1, including $\alpha$-Precision, $\beta$-Recall, and Detection Score.
> >
> > **If by “data fidelity” you referring to Machine Learning Efficiency (MLE).** We acknowledge that the results in our initial manuscript showed that TabSyn performs slightly better on average, which is rather unexpected given that TabSyn underperforms TabDiff on all data fidelity metrics.
> >
> > After re-examing our MLE experiments setup, we discovered an implementation mistake in the MLE evaluation for TabSyn. Specifically, when validating the classifier trained on data generative by the models, we applied a different validation criterion for TabSyn’s trials compared to the other models. After correcting this inconsistency, we reran the experiments and found that TabSyn’s performance on MLE decreased slightly. With this correction, TabDiff now outperforms TabSyn on average in MLE as well.
> >
> > The updated MLE results have been included in the revised version of the paper (Table 3). The corrected results further support that TabDiff can generate more authentic data that better capture the statistical features of the original data distribution, thus better empowering the learning of downstream tasks.
> >
> > >**[W3] Meaning of “multimodal” in the title**
> >
> > Thank you for your comment regarding the title and the interpretation of “multimodal”. We understand that it might seem a bit confusing, as also suggested by Reviewer UD32.
> >
> > **What we mean by “multimodal”.** Yes, by “multimodal”, we refer to the fact that in tabular data, different columns contain various data types. And yes, under the scope of the problem we solve, those data types only include numerical and categorical table entries. We use this term in order to highlight the challenge of building a unified generative framework that handles the distinct characteristics of these different data types.
> >
> > We understand the confusion this term entails, as this term typically describes data types with more salient distinctions, like image vs. text. We initially thought that this term could help us clearly highlight the mixed-type nature of tabular data. We appreciate your question and we are open to refining the title to better convey this intended meaning.
> >
> > ---
> > Hope our response can address your questions!

---

> > > ### Comment · Reviewer_Buu2 · 2024-11-20
> > >
> > > Thank you for addressing my concerns. I'll update my score to 6.

---

> > > > ### Author Response · Authors · 2024-11-20
> > > > **Thank you very much for your feedback!**
> > > >
> > > > Thank you for your quick reply! And thank you very much for providing valuable feedback and recognizing our efforts!

---

> > > > > ### Comment · Reviewer_Buu2 · 2024-11-25
> > > > >
> > > > > Dear authors,
> > > > >
> > > > > I noticed a public comment mentioning another paper that also utilizes an adaptive learnable noise schedule. However, this paper is not cited in your manuscript. Could you please clarify the relationship and differences between your work and the paper mentioned in the public comment? Additionally, could you explain the rationale behind not citing this work?
> > > > >
> > > > > Looking forward to your clarification.

---

> > > > > > ### Author Response · Authors · 2024-11-27
> > > > > > **Response to related work**
> > > > > >
> > > > > > Thank you for bringing this to our attention. We carefully read through the paper and the author’s questions, and are pleased to clarify the relationship and differences between TabDiff and Continuous Diffusion for Mixed-Type Tabular Data (CDTD) [1].
> > > > > >
> > > > > > > **Relations and differences**
> > > > > >
> > > > > > TabDiff and CDTD both use continuous-time diffusion models and introduce learnable noise schedules to handle heterogeneity across various features in tabular data. However, the diffusion framework and design are fundamentally different and come from different motivations, which we elaborate on as follows:
> > > > > >
> > > > > > 1. **Different modeling of discrete features.** TabDiff differs significantly from CDTD in how it models the discrete variables (categorical features). Unlike CDTD which encodes discrete features into continuous latent space and requires quantization during sampling, our approach models numerical and categorical features separately in their natural spaces. By directly modeling categorical features through a masking process rather than continuous latent vectors, we better preserve their inherent discrete nature.
> > > > > >
> > > > > > 2. **Different motivation and design for featurewise noise schedule.** While TabDiff and CDTD use adaptive learnable noise schedules to handle heterogeneity across various features in tabular data, their motivations differ. CDTD is motivated by the observation that simpler features (e.g. a binary categorical feature) need more noise than complex ones in order to achieve equal signal levels across all features. Thus, an adaptive noise schedule plays an important role in synchronizing the signal levels across features. In contrast, TabDiff's adaptive noise scheduling is motivated by the desire to mimic a sequential generation process --- allowing features generated slower in the process to be conditioned on those generated faster. While video diffusion models [2] have shown different noise schedules can create ordered diffusion, tabular data lacks video’s temporal ordering. Inspired by the Variational Diffusion Models (VDM) [3] paper, we, therefore, let the model learn its own generation order through learnable noise schedule parameters. Besides the different motivations, the specific designs of noise schedules also differ. CDTD employs flexible S-shaped curves for its noise schedules in the continuous space, while TabDiff simply adjusts the concaveness of the original schedules (as illustrated in [the response to [W2] of Reviewer zTT5](https://openreview.net/forum?id=swvURjrt8z&noteId=ldtIzlb5tH)), which are sufficient to establish generation order.
> > > > > >
> > > > > > > **Missing references**
> > > > > >
> > > > > > The omission of CDTD from related work and comparison is due to its concurrent nature and use of different benchmarks whose implementation hasn’t been publicly available. However, given its valuable insights, we have updated our manuscript to cite CDTD and further clarify its relationship to and distinctions with TabDiff.
> > > > > >
> > > > > > [1] Mueller, et al., 2024. "Continuous Diffusion for Mixed-Type Tabular Data."
> > > > > >
> > > > > > [2] Ruhe, et al., 2024. “Rolling Diffusion Models.”
> > > > > >
> > > > > > [3] Kingma et al., 2021. “Variational Diffusion Models.”
> > > > > >
> > > > > > ---
> > > > > > We hope this addresses your questions!

---

> > > > > > > ### Comment · Reviewer_Buu2 · 2024-11-27
> > > > > > >
> > > > > > > Thank you for your clarification, which resolved my questions.

---

### Official Review · Reviewer_kzHM · 2024-11-01

**Soundness:** 3
**Presentation:** 3
**Contribution:** 3
**Rating:** 6
**Confidence:** 3

**Summary:**

The paper introduces TabDiff, an innovative multi-modal diffusion model designed to generate high-quality synthetic tabular data. It addresses the challenges of heterogeneous tabular data, which includes both numerical and categorical features with complex interdependencies. TabDiff employs a joint continuous-time diffusion process with feature-wise learnable components, allowing it to handle different data modalities while adapting to the varying distributions of individual features.

**Strengths:**

1. Tabdiff introduces a unified joint continuous-time diffusion process with feature-wise learnable components for handling both numerical and categorical data is innovative and allows the model to adapt to the specific characteristics of different data distributions.
2. The paper provides a thorough evaluation of TabDiff on a wide range of metrics, offering a deep and detailed assessment of the model performance from multiple dimensions, which is highly commendable.
3. The paper emphasizes the importance of TabDiff in generating high-quality synthetic data that enables effective model training while ensuring privacy preservation. The authors validate this effectiveness through the use of various metrics, demonstrating that the model not only maintains data fidelity but also safeguards against privacy risks. It's a practical trial for addressing the growing need for privacy-conscious data generation in sensitive domains.

**Weaknesses:**

1. Although the paper does not explicitly highlight it, the complexity of TabDiff, particularly in handling high-dimensional features through diffusion, could result in increased computational costs compared to simpler models. It would be good to provide a comparison of generation or training speed metrics with other baselines to better illustrate this trade-off.

2. The performance drop of TabDDPM on the News and Diabetes datasets is surprising, especially given that other diffusion-based methods maintain stable results. This discrepancy is beyond expected boundaries, with precision and recall even dropping to 0 in Table 7 and Table 8, and similarly poor performance across other tables for these datasets.
Thus, while I appreciate the authors' comprehensive comparison across various metrics with baselines, the paper lacks sufficient analysis and discussion of the experimental results, particularly in explaining the performance gaps between TabDiff and the baselines. It would strengthen the paper if the authors provided more detailed discussion, and offered insights into the experimental setup of the baselines.

3. The paper lacks sufficient explanation of the experimental setup for these baselines, which makes it harder to interpret the results. Providing more clarity on these aspects would strengthen the case for TabDiff's performance and offer a more comprehensive understanding of the comparative results.

**Questions:**

See weaknesses, I will raise the score if all the concerns are addressed.

---

> ### Author Response · Authors · 2024-11-20
> **Author Response Part 1**
>
> Thank you for your constructive feedback and insightful questions! Below, we provide detailed responses to each of your questions. Additionally, please refer to our revised paper, with updates colored in purple.
>
> >**[W1] Model Efficiency**
>
> This is an excellent suggestion, and we totally agree that a thorough analysis of model efficiency could further enhance our methods’ evaluation. Below, we first compare the training and sampling speed of TabDiff against other competitive baseline methods (TabDDPM, TabSyn) using **four different metrics**. Additionally, inspired by your suggestion, we conduct a new ablation study exploring the tradeoff between discretization steps (i.e.,  efficiency) and sample quality in diffusion-based models. We use the representative Adult dataset, which contains a balanced number of numerical and categorical columns, throughout the experiment Our results show that, among the competitive methods, TabDiff is not only the most effective but also highly efficient. TabDiff also maintains the most capacity when the sampling discretization steps are limited. Below, we present detailed analyses.
>
>
> **Metric#1: Training Time.** We first measure the training time of each method. The epoch lengths are set to the default configuration of each method, and the validation frequencies are set to the same value of once every 200 epochs. Our measurements are presented in the first column of the table below, with the entry of TabSyn being split into – VAE training time + Diffusion training time.
>
> We see that all three methods take a comparable time to complete one training run, with Tabsyn being $10\%$ faster than TabDiff and TabDiff being $20\%$ faster than TabDDPM. The current gap between TabDiff and TabSyn is likely due to TabDiff’s slightly deeper network architecture compared to TabSyn (note: deeper but not larger, as we enforce the same parameter size for all models in our experiments). We believe that the architecture of TabDiff can be further optimized to improve efficiency, and we leave this as future work.
>
> Nevertheless, it is also important to consider model robustness when assessing efficiency. As highlighted in our response to [W2] raised by Reviewer gKXC, the training process for TabSyn is notably unstable due to the difficulty in training VAEs, often requiring you to retrain many times in order to produce a model capable of generating samples comparable in quality to TabDiff. On the other hand, TabDDPM fails to scale to more complicated datasets as mentioned in the response to [W2] you raised. Thus, when taking into account training robustness, TabDiff is the most robust and efficient model among all competitive methods.
>
> **Metric#2: Training Convergence.** Next, we assess training convergence by evaluating the quality of samples generated from intermediate checkpoints during the training process. Figure 6 in the revised manuscript plots our result. The curves show that TabDiff converges faster than the other methods, as TabDiff produces more high-quality samples when shown to the datasets for the same number of times (i.e., same epoch).
>
> **Metric#3: Number of Function Evaluation (NFE).** For sampling, we first theoretically analyze the number of network calls involved in a single diffusion step (i.e. the denoising step from $x_t$ to $x_{t-1}$)) for each method. The numbers are presented in the third column of the table below. TabDiff and TabSyn involve an extra network call because TabDiff employs the second-order correction trick introduced in [1].
>
> **Metric#4: Sampling Time.** We measure empirically the time to generate the same number of samples as the test set (32561 samples for Adult). The numbers are presented in the second column of the table below. According to it, TabDiff and TabSyn samples $\sim10\times$ faster than TabDDPM. This result is expected, as both TabDiff and TabSyn, by default, use 20 times fewer sampling steps than TabDDPM, while making twice as many network calls per step compared to TabDDPM. TabDiff’s slightly longer sampling time is also attributed to the evaluation of its deeper network. We believe this can be optimized in future work.
>
> | Method          | Training Time (min) | Sampling Time (sec)   | NFE |
> |------------------|---------------------|------------------------|-----|
> | TabDDPM         | 112                 | $125.1 \pm 0.01$      | 1   |
> | TabSyn          | 45 + 40 = 85        | $8.8 \pm 0.005$       | 2   |
> | TabDiff (ours)  | 94                  | $15.2 \pm 0.007$      | 2   |

---

> > ### Author Response · Authors · 2024-11-20
> > **Part 1 (Continue)**
> >
> > **Control quality-efficiency tradeoff through discretization steps.** One advantage of continuous-time diffusion models (which includes TabDiff and TabSyn under our task) is the ability to sample with arbitrary discretization steps, allowing them to flexibly tradeoff sample quality with sample efficiency. We conduct an experiment that compares how TabDiff and TabSyn perform when sampled with different sample steps. The results are presented in the table below, and a visualization is presented in Figure 7 of the revised paper.
> >
> > | Steps | **TabSyn Shape** | **TabSyn Trend** | **TabDiff Shape** | **TabDiff Trend** |
> > |-------|-------------------|------------------|--------------------|-------------------|
> > | 5     | 34.09            | 49.30           | 12.51             | 22.15            |
> > | 10    | 1.99             | 3.92            | 1.55              | 3.36             |
> > | 25    | 0.84             | 1.96            | 0.62              | 1.50             |
> > | 50    | 0.81             | 1.95            | 0.63              | 1.49             |
> > | 100   | 0.82             | 1.94            | 0.64              | 1.53             |
> >
> > Our results show that TabDiff consistently achieves higher sample quality across all levels of discretization steps. Notably, when the number of steps is reduced to just 5 (requiring only one second for sampling), TabSyn fails to generate meaningful content, as indicated by an error rate approaching $50\%$. In contrast, TabDiff continues to produce valid and coherent content even under these highly constrained conditions.
> >
> > [1] Karras et al., 2022. “Elucidating the Design Space of Diffusion-Based Generative Models.”

---

> > ### Author Response · Authors · 2024-11-20
> > **Author Response Part 2**
> >
> > > [W2, W3] **Baseline model TabDDPM’s performance drops + detailed experiment setup and result interpretation**
> >
> > **TabDDPM’s performance drops.** This is a very good observation, and we were also surprised when we received the results. As mentioned in Appendix D, the poor performance of TabDDPM on the larger datasets News and Diabetes is because TabDDPM fails to generate meaningful samples or cover the diversity of data distribution. For instance, as we can see in Figure 11 of the revised manuscript,  the numerical columns of TabDDPM’s Diabetes samples all collapse to the minimal and maximal values of the domains.
> >
> > After examining the training logs, we discovered that this poor generation performance might be due to the explosion of training loss, as shown in Figure 10 of the revised manuscript.
> >
> >
> > News and Diabetes are notably larger than the rest of the datasets: News contains the most number of features, while Diabetes has the most number of rows and contains a few high-dimensional categorical features. Given this, we can infer that TabDDPM doesn’t scale as well as other diffusion-based models like TabSyn and our method.
> >
> > **Why TabDiff shows much more consistent performance.** TabDDPM is fairly close to our method while our method scales perfectly to those large datasets. Thus, we naturally speculate this distinction is due to a few design benefits of our method.
> > - First, with respect to the diffusion modeling of numerical features, the EDM parameterization employed by our method, as well as TabSyn, is more robust than the original discrete-time DDPM parameterization [1].
> > - Second, regarding the diffusion modeling of categorical features, TabDiff employs a simple masked diffusion framework that is easier and more stable to train and optimize compared to the general discrete diffusion framework used by TabDDPM. In the general diffusion framework, the training objective is evaluated across all columns. Whereas,  TabDiff’s masked diffusion framework solely computes the loss on columns that have been masked [Eq.(9)].  Moreover, our loss is weighted by the simple factor of $\frac{\alpha_t^{\prime}}{1-\alpha_t}$, which simplifies to $\frac{-k}{1-\alpha_t}$ under our noise schedule parameterization.
> > - Finally, TabDiff introduces adaptive learnable noise schedules that better tackle the feature-wise heterogeneity within tabular data and allow the model to better capture intercolumn correlations, as evidenced by TabDiff's superior performance on Trend shown in Table 2.
> >
> >
> > **Details about experiment setup.** We appreciate your suggestion and present a more detailed discussion on it in the following:
> >
> > First, regarding the preprocessing of tabular data, all methods in the paper employ the same preprocessing procedure described in Appendix D., and this procedure is directly followed from the convention in [2, 3]. All methods are experimented with the same dataset split described in Appendix D.
> >
> > The implementations of baseline methods are borrowed directly from the codebase compiled by [3]. These implementations are either taken directly from the official repository of the official repositories of the respective method or modified to support mixed-type tables. To ensure a fair comparison, the network sizes of all baseline methods are standardized to ~10M parameters, with some older baseline methods being scaled up accordingly. Further details on adaptations and network scaling can be found in Appendix G of [tabsyn]. Additionally, when training on the newly introduced datasets Diabetes, we retain each baseline model’s original hyperparameters, such as training epoch and optimization details, without modification.
> >
> > The implementation details of our method TabDiff, including architecture designs and hyperparameter settings, are thoroughly discussed in Appendix D. The network size is set to be around ~10M accordingly.
> >
> > Details of the evaluation metrics and evaluation processes are provided in Appendix A, largely following those in [3].
> >
> > Lastly, we plan to *release our code at the end of the discussion period*!
> >
> > **Detailed analysis of experiment results.** The basic analysis of all our experiment results is either presented in Section 4 or deferred to Appendix D due to space limits. Additional experiments and analyses (e.g. the study of models’ efficiency and robustness) inspired by your and other reviewers’ comments are **highlighted in purple in the revised version of the manuscript**.
> >
> > [1] Karras et al., 2022. “Elucidating the Design Space of Diffusion-Based Generative Models.”
> >
> > [2] Kotelnikov et al., 2023 “TabDDPM: Modelling Tabular Data with Diffusion Models.”
> >
> > [3] Zhang et al., 2024 “Mixed-Type Tabular Data Synthesis with Score-based Diffusion in Latent Space.”
> >
> > ---
> > Hope our response can address your questions!

---

> ### Author Response · Authors · 2024-11-24
> **Looking forward to feedback during the discussion period**
>
> Dear Reviewer kzHM,
>
> We would like to express our gratitude for your valuable comments and suggestions. Thanks a lot for your positive feedback on our method and experiments.
>
> As the author-reviewer discussion period **is coming to a close on Nov 27th**, we would greatly appreciate it if you could clarify any remaining concerns you may have. This will ensure that we can adequately address them during this period.
>
> Thank you very much for your attention to this matter.
>
> Best regards,
>
> Authors

---

> > ### Comment · Reviewer_kzHM · 2024-11-26
> >
> > Thanks for your efforts. My initial concerns have been addressed.

---

> > > ### Author Response · Authors · 2024-11-27
> > > **Thank you very much for your feedback!**
> > >
> > > Thank you very much for providing valuable feedback and recognizing our efforts!

---

### Official Review · Reviewer_UD32 · 2024-11-02

**Soundness:** 3
**Presentation:** 3
**Contribution:** 3
**Rating:** 6
**Confidence:** 3

**Summary:**

This paper introduces an approach called TABDIFF, which develops a joint continuous-time diffusion process for mixed-type data in tabular datasets. The authors address the challenge of high disparity between different feature distributions by proposing feature-wise learnable diffusion processes. The key contribution is a hybrid diffusion process to handle numerical and categorical features.

**Strengths:**

**1** The experiments are extensive, demonstrating the efficiency of the proposed method.

**2** Hybrid diffusion models are employed to handle mixed-type data in tabular datasets.

**Weaknesses:**

**1** The authors claim that they explicitly tackle the feature-wise heterogeneity issue in the multi-modal diffusion process. However, the details about how to handle this problem is missing.

**2** Even though experiments show the efficiency of the proposed method, it lacks of the motivation of considering a masked diffusion for categorical features.

**Questions:**

**Q1** The title of this paper is somewhat confusing. Before reading it, I assumed it referred to a diffusion model capable of generating various types of data structures, such as tabular data and images. After reading, I believe the term "modality" may refer to either of two scenarios: (1) multiple modalities within a specific variable (a column in a tabular dataset); (2) columns of a tabular dataset that are mixed-type. If the latter is the intended meaning, the paper should demonstrate how the proposed method differs from TabDDPM, as TabDDPM also handles mixed-type data generation in tabular datasets. It is inappropriate to use such an ambitious title.


**Q2** The proposed method utilizes the masked diffusion for categorical features. What is the point of using masked diffusion?

---

> ### Author Response · Authors · 2024-11-20
> **Author Response Part 1**
>
> Thank you for your constructive feedback and insightful questions! Below, we provide detailed responses to each of your questions. Additionally, please refer to our revised paper, with updates colored in purple.
>
> > **[W1] How TabDiff tackles feature-wise heterogeneity**
>
> This is a very good point. We provide a more detailed discussion in the following:
>
> **Recap: what do we mean by feature-wise heterogeneity.** By feature-wise heterogeneity, we refer to the fact that, in tabular data, each feature/column often exhibits marginal distribution in distinct shapes (which is illustrated by the blue density curves in Figure 11 of the revised paper). This poses an extra challenge for generative modeling: columns with more complex marginals are more difficult to model, so generative models should pay more attention to them.
>
> **How TabDiff handles this problem.** In our approach, we address this challenge by implementing *adaptive learnable noise schedules*, as more thoroughly discussed in Section 2.3. These schedules allow the model to adaptively adjust the diffusion process for each feature throughout the training process. This, in turn, eventually lets the model optimally allocate its capacity to capture distinct distributions of different columns, reducing learning bias towards any specific column.
>
> **Evidence of the improvement.** In our original submission, we demonstrated the effectiveness of this approach in our ablation study (Section 4.4, Table 5). Specifically, TabDiff with learnable schedules achieves a lower average error of column-wise density (i.e. Shape) across columns compared to a version with uniform schedules, indicating a better ability to handle heterogeneous feature distributions.
>
> **Addressing the Lack of Detailed Explanation.** We totally agree with your point that we need to address what we mean by “optimal allocation of model capacity across heterogeneous features” in greater detail. Inspired by your feedback, we further study the impact of learnable schedules by **analyzing how generation quality varies between different features**. The Shape and Trend errors presented in Table 5 are averaged over all columns and columns pars. Now, in the revised manuscript, using the representative Adult dataset,  we visualize the errors at each column and column pair in Figures 8 and 9, along with the normalized standard deviations in the table below to quantitatively measure the variations of errors. The results show that TabDiff with learnable schedules not only achieves lower average error **but also exhibits more consistent errors across columns**. This balance indicates that learnable schedules help the model balance its capacity on different dimensions of the data, improving the model’s ability to handle heterogeneous distributions.
>
> | **Method**          | **Shape** | **Trend** | **Shape Std.** | **Trend Std.** |
> |----------------------|-----------|-----------|----------------|----------------|
> | TabSyn             | 0.81      | 1.93      | 1.01           | 0.88           |
> | TabDiff-Fixed       | 0.74      | 1.73      | 0.42           | 0.75           |
> | TabDiff-Learn      | **0.63**  | **1.49**  | **0.29**       | **0.64**       |

---

> ### Author Response · Authors · 2024-11-20
> **Author Response Part 2**
>
> >**[W2, Q2] Motivation of masked diffusion**
> This is another good point. Our choice is primarily motivated by the following advantages that masked diffusion offers over other discrete diffusion frameworks.
>
> **1. Better numerical stability and efficiency in training.** The masked diffusion framework has a very simple objective. Unlike more generalized discrete diffusion frameworks [1, 2], where the loss is evaluated across all columns, our framework computes the loss solely on columns that have been masked [Eq. (9)]. Moreover, our loss is weighted by the simple factor of $\frac{\alpha_t^{\prime}}{1-\alpha_t}$, which simplifies to $\frac{-k}{1-\alpha_t}$ under our noise schedule parameterization. This simple training objective enhances both training numerical stability and efficiency [3, 4].
>
> These benefits were initially demonstrated in experiments on texts [3, 4]. Recognizing that categorical columns of tabular data can be considered sequences of unordered tokens similar to text, we were motivated to apply the simple masked diffusion framework to tabular data generation. As expected, we observed similar benefits of stable training.
>
> To illustrate these, we draw a direct comparison with TabDDPM, which utilizes a general discrete diffusion framework that interpolates between data and a uniform prior. TabDiff shows more stable training when scaling up to datasets with more complex categorical columns. For instance, on the Diabetes datasets which include multiple categorical features with very large numbers of classes (Table 6), the training of TabDDPM failed to converge properly, as evidenced by the exploding training loss in Figure 10 of the revised paper. Consequently, TabDDPM failed to generate meaningful content, resulting in poor Shape and Trend scores as presented in Tables 1 and 2. In contrast, TabDiff, utilizing the simple masked diffusion framework, exhibits very stable training across all datasets and consistently produces high-quality samples. Furthermore, TabDiff is more efficient to train than TabDDPM, as shown by its shorter training time and faster convergence in Table 11 and Figure 6 of the revised paper.
>
> **2. Compatibility with Auxiliary Techniques.** The second advantage is that the simple masked diffusion framework is easily compatible with auxiliary techniques [4]. In our paper, we successfully adopted the classifier-free guidance technique for conditional generation, in a fairly clean matter.  In the Appendix B.2 revised version of the paper, we have shown that the CFG guided posterior probability $\tilde{p}\_{\theta}(x\_s^{\text{cat}} | x\_t, y)$ can be simply computed by directly interpolating the model's raw estimates of $x_0$, i.e.,$\mu_{\theta}^{\text{cat}}(x_t, y, t)$ and $\mu_{\theta}^{\text{cat}}(x_t, t)$.
>
> [1] Austin et al., 2021. “Structured Denoising Diffusion Models in Discrete State-Spaces.”
>
> [2] Campbel et al., 2022 “A Continuous Time Framework for Discrete Denoising Models.”
>
> [3] Lou et al., 2024, “Discrete Diffusion Modeling by Estimating the Ratios of the Data Distribution.”
>
> [4] Sahoo et al., 2024, “Simple and effective masked diffusion language models.”

---

> > ### Author Response · Authors · 2024-11-20
> > **Author Response Part 3**
> >
> > >**[q1] Meaning of “multimodal” in the title**
> > Thank you for your comment regarding the title and the interpretation of “multimodal”. We totally understand it and have anticipated the potential confusion when drafting the title.
> >
> > **What we mean by “multimodal”.** By “multimodal”, we intend to refer to the second scenario you described: different columns contain various data types. Under the scope of the problem we solve, these data types include 1-D numerical and categorical table entries. We use this term in order to highlight the challenge of building a unified generative framework that handles the distinct characteristics of these different data types.
> >
> > **Comparison with TabDDPM.** Yes, under this definition of “multimodal”, TabDDPM, along with other diffusion-based baselines such as TabSyn and CoDi, can also be regarded as multimodal diffusion models, for they indeed model the same type of mixed-type tabular as TabDiff. Nevertheless, the way TabDDPM and TabDiff model the multimodal distribution differs significantly. First, the unified diffusion processes in TabDiff are modeled in continuous time, allowing it to flexibly trade-off sample quality and efficiency by training just one model (and sampling at arbitrary numbers of discretization steps). Second, in comparison to the general discrete diffusion framework employed by TabDDPM for categorical data, TabDiff employs a simple masked diffusion process that is not only easier to understand but also demonstrates more stable and efficient training as mentioned in the response to Q2. Finally, TabDiff introduces adaptive learnable noise schedules that better tackle the feature-wise heterogeneity within tabular data, and this boosts sample quality as discussed in the response to W1. These design improvements make TabDiff more competent in modeling multimodal tabular data.
> >
> > **Claiming “multimodal” is a bit ambitious.** We totally understand the confusion this term entails, as this term typically describes data types with more salient distinctions, like image vs. text. We initially thought that this term could help us clearly highlight the mixed-type nature of tabular data. We appreciate your suggestions and we are open to refining the title to better convey this intended meaning.
> >
> > ---
> > Hope our response can address your questions!

---

> > > ### Comment · Reviewer_UD32 · 2024-11-21
> > > **Response to authors**
> > >
> > > Thanks for clarification. I will raise my score to 6.

---

> > > > ### Author Response · Authors · 2024-11-21
> > > > **Thank you very much for your feedback!**
> > > >
> > > > Thank you for your quick reply! And thank you so much for providing valuable feedback and recognizing our efforts!

---

### Official Review · Reviewer_zTT5 · 2024-11-02

**Soundness:** 3
**Presentation:** 3
**Contribution:** 3
**Rating:** 6
**Confidence:** 3

**Summary:**

This work proposed a multi-modal diffusion framework for generating high-quality synthetic data. A key innovation is the development of a joint continuous-time diffusion process for numerical and categorical data, where authors propose learnable column-wise noise schedules to counter the high disparity of different feature distributions. Extensive experiments are conducted to demonstrate the superior capacity of proposed approach.

**Strengths:**

This manuscript demonstrates advantages in multiple aspects. It provides a clear and comprehensive introduction of the research problem, and conducts extensive experiments on multiple data sets and evaluation criteria.

Additionally, the paper elaborates on relevant algorithms and techniques. The algorithmic sections present some innovative and original findings, contributing to advancements in the field of machine learning and offering new insights to the academic community.

**Weaknesses:**

1. Some necessary explanations are lacking in the description of algorithm provided in the Method and Appendix, requiring more detailed descriptions to enhance the readability of the paper.

2. The description of categorical column needs an example to illustrate the method more clearly.

3. Formula (10) is missing a parenthesis.

4. Why are formula (10) and formula (11) constructed this way? Is there a source? If not, can you explain the construction idea?

**Questions:**

See weakness.

---

> ### Author Response · Authors · 2024-11-20
> **Author Response Part 1**
>
> Thank you for your constructive feedback and insightful questions! Below, we provide detailed responses to each of your questions. Additionally, please refer to our revised paper, with updates colored in purple.
>
> >**[W1, W2] Detailed illustration for TabDiff’s diffusion processes and the modeling of categorical columns**
>
> Thank you for pointing this out. We completely agree that the presentation of our model’s training and sampling process would benefit from an intuitive illustration.
>
> Now in Appendix C (“Detailed Illustrations of Training and Sampling Processes”) of the revised manuscript, we closely illustrate the diffusion training and testing processes **using an intuitive visualization in Figure 5** that showcases the process of generating a toy example. We also more closely described the modeling of categorical columns by referring to specific examples in this illustration.
>
> >**[W3] Missing parenthesis in Eq.(10)**
>
> Thank you for pointing this out. We have corrected it in the revised manuscript!

---

> > ### Author Response · Authors · 2024-11-20
> > **Author Response Part 2**
> >
> > > **[W4] Motivation for noise schedules**
> >
> > We sincerely appreciate your insightful comments regarding our description of noise schedules. Based on your feedback, we have provided a more detailed explanation of our motivation for selecting power-mean and log-linear schedulers, and have conducted additional experiments to validate these choices.
> >
> > **Background.** The power-mean and log-linear schedulers are well-established noise schedules in diffusion and related applications.
> >
> > **Power-mean Scheduler.** Our power mean schedule for numerical features is inspired by the seminal work [1]. It is formulated as:
> >
> > $$
> > \sigma_{i<N} = \left( \sigma_{\max}^{\frac{1}{\rho}} + \frac{i}{N-1} \left( \sigma_{\min}^{\frac{1}{\rho}} - \sigma_{\max}^{\frac{1}{\rho}} \right) \right)^{\rho}, \quad \sigma_N = 0
> > $$
> >
> > where $t = \frac{i}{N-1}$ in continuous time.
> >
> > In essence, this schedule is nothing but a polynomial function from 0 to 1 with the maximum and minimum fixed to the predetermined values $\sigma_{\min}$ and $\sigma_{\max}$, as we can see that when $\rho=1$, the function degenerates to a straight line. A $\rho > 1$ would imply that the model puts more focus on the diffusion steps closer to the clear data, whereas a $\rho < 1$ would imply that the model puts more focus on the diffusion steps closer to the noise. In our implementation, we enforce the restriction $\rho > 1$, based on the empirical finding of [1] that allocating more resources to learn the denoising step close to clear data gives better performance.
> >
> > To design our feature-wise learnable scheduler, we thus directly replace $\rho$ with a learnable parameter $\rho_i$ for each continuous feature $i$.
> >
> > **Log-linear Scheduler**
> >
> > In discrete-space diffusion, the log-linear noise schedule function has been widely adopted across multiple studies [2, 3, 4, 5].
> >
> > The log-linear scheduler is formulated as $\sigma(t) = -\log(1 - t)$. With this scheduling, the relationship between the signal level $\alpha$ and time $t$ becomes
> >
> > $$
> > \alpha(t) = 1 - t ,
> > $$
> >
> > which implies the intuitive meaning that the signal level linearly decays from $t=0$ to $t=1$.
> >
> > The recent work [4] demonstrated this function's superior performance in terms of likelihood (bits per dimension) compared to alternatives such as cosine and linear schedulers.
> >
> > In our implementation, we enhanced this function by introducing a learnable power term $k_j$ for each discrete feature $j$ as $\sigma(t) = -\log(1 - t^{k_j})$.
> >
> > **Validating with new experiments.** Inspired by your question, we investigated an alternative approach using a more flexible architecture. A legitimate noise schedule only needs to ensure that it monotonically increases with the timestep, as data get more noisy in the forward diffusion process. Thus, we designed a larger class of such learnable noisy schedules, parameterized by a 2-layer monotonically increasing feature-wise MLP:
> >
> > $$
> > \sigma_i(t) = \text{MLP}_i(t), \text{ for }i\text{th feature in the table}
> > $$
> >
> > Our hypothesis was that the highly flexible MLP might lead to overfitting and be hard to optimize, thereby reducing TabDiff’s robustness. To test this, we conducted comprehensive experiments comparing various scheduler configurations across all datasets. The results are shown in the following table, with P-m and L-l standing for Power-mean and Log-linear, respectively.
> >
> > | Scheduler | Shape | Trend |
> > |---------------|-------|-------|
> > | MLP | 1.67 | 4.43 |
> > | P-m(fixed) + L-l (fixed) | 1.20 | 1.93 |
> > | P-m (fixed) + L-l (trainable) | 1.33 | 1.89 |
> > | P-m (learned) + L-l (fixed) | 1.24 | 1.81 |
> > | P-m (learned) + L-l (leared) | **1.17** | **1.80** |
> >
> > These results validate our initial assumptions and demonstrate the effectiveness of our approach in making noise schedulers learnable while maintaining robustness. The significantly higher error rates in the MLP configuration confirm that excessive flexibility in the scheduler can indeed lead to degraded performance.
> >
> > [1] Karras et al., 2022. “Elucidating the Design Space of Diffusion-Based Generative Models.”
> >
> > [2] Austin et al., 2021. “Structured Denoising Diffusion Models in Discrete State-Spaces.”
> >
> > [3] Lou et al., 2024, “Discrete Diffusion Modeling by Estimating the Ratios of the Data Distribution.”
> >
> > [4] Sahoo et al., 2024 “Simple and effective masked diffusion language models.”
> >
> > [5] Sohl-Dickstein et al., 2025. "Deep Unsupervised Learning Using Nonequilibrium Thermodynamics."
> >
> > ---
> > Hope our response can address your questions!

---

> > > ### Comment · Reviewer_zTT5 · 2024-11-25
> > > **Thanks for your response**
> > >
> > > Thanks for your responses and clarifications. I'll keep my score.

---

> > > > ### Author Response · Authors · 2024-11-25
> > > > **Thank you very much for your feedback!**
> > > >
> > > > Thank you very much for providing valuable feedback and recognizing our efforts!

---

> ### Author Response · Authors · 2024-11-24
> **Looking forward to feedback during the discussion period**
>
> Dear Reviewer zTT5,
>
> We would like to express our gratitude for your valuable comments and suggestions. Thanks a lot for your positive feedback on our method and experiments.
>
> As the author-reviewer discussion period **is coming to a close on Nov 27th**, we would greatly appreciate it if you could clarify any remaining concerns you may have. This will ensure that we can adequately address them during this period.
>
> Thank you very much for your attention to this matter.
>
> Best regards,
>
> Authors

---

### Official Review · Reviewer_gKXC · 2024-11-04

**Soundness:** 4
**Presentation:** 4
**Contribution:** 3
**Rating:** 6
**Confidence:** 3

**Summary:**

While extensive work has attempted to solve the task of tabular data generation, they often face different problems, such as the failure of cross-modal modeling and the additional cost of training auto-encoders. To this end, this paper proposes TabDiff, a diffusion-based model that unifies modal heterogeneity within one single approach. The proposed method, built upon several existing techniques, models numerical and categorical attributes separately and introduces a learnable noise scheduler that further facilitates the sampling process. Extensive experiments verify the effectiveness of TabDiff. Ablation studies also provide additional insights.

**Strengths:**

1. The paper is well-written and well-structured. It is self-content and easy to follow.
2. The problem investigated in this paper is well-motivated and practical.
3. While most of the techniques have been investigated in prior work, this work seamlessly integrates them and adapts to the tabular data generation task.
4. Extensive experiments sufficiently verify the effectiveness of the proposed approach, and ablation studies further outline the design choices and the benefits of the learnable noise schedulers.

**Weaknesses:**

1. Despite the claim of cross-modal modeling, the proposed method currently still models numerical and categorical values separately, which might make it difficult to synthesize more complicated tabular datasets.
2. While I generally appreciate the effort and the performance, I am not fully convinced by the claim of the multi-modal diffusion framework. The experiments in Tables 1, 2, and 3 all suggest comparable performance compared to TabSyn. As TabSyn has been published in ICLR 2024, the authors should clearly state the main difference and closely compare the two works.
3. Lack of motivation for schedulers. What is the motivation for choosing power-mean and log-linear schedulers? The choice looks random to me.

**Questions:**

1. Do numerical and categorical value modeling help each other, or are they just modeled separately?

---

> ### Author Response · Authors · 2024-11-20
> **Author Response Part 1**
>
> Thank you for your constructive feedback and insightful questions! Below, we provide detailed responses to each of your questions. Additionally, please refer to our revised paper, with updates colored in purple.
>
> >**[W1, Q1] Cross-modal modeling**
>
> Thank you for raising the point regarding how TabDiff models numerical and categorical features together. We elaborate on this point in the following.
>
> **Are the features modeled separately?**  First, we would like to clarify that, TabDiff does NOT model different types of features separately, though it uses different diffusion processes for them. TabDiff models the joint of all features under the same continuous time diffusion framework, where all features are perturbed with noise together in the forward training process and denoised together in the reverse sampling process. The diffusion process is designed differently in order to accommodate the distinct natures of each data type (e.g. we destroy the discrete nature of categorical features if we add gaussian noise to them).
>
> **How modelings of different features help each other.** The modelings of numerical and categorical features help each other by capturing their mutual dependencies within the joint distribution. Such dependencies are captured in TabDiff in two aspects.
>
> First, in the reverse diffusion process, we adopt the common practice in multimodal diffusion models by factorizing the posterior across different features, as described in Eq.(2). This factorization does not imply that the features are independent. Each conditional distribution $q(x_s^{\text{num}} \mid x_t, x_0)$  and $q(x_s^{\text{cat}} \mid x_t, x_0)$ is conditioned on the entire $x_t$ and $x_0$, which contains information from both numerical and categorical features. This allows the model to learn and capture complex dependencies between different features.
>
> Beyond this standard practice, the modeling of dependencies between different features is enhanced by the feature-wise adaptive noise schedules. These learnable schedules give the model the flexibility to denoise different features at varying speeds. Specifically, a more positive $\rho$ and $k$ corresponds to quicker denoising. During training, the model learns different denoising rates for different features feature, which can be regarded as an order of generation. Features with larger $\rho$ and $k$ are recovered earlier in the denoising process, allowing subsequent features to be generated and conditioned on them. This mechanism enables the model to better capture inter–feature dependencies. In Table 2, TabDiff outperforms all other baselines on inter-column correlation metrics (i.e., Trend); in Table 5, TabDiff with learnable schedules achieves higher scores on Trend.

---

> > ### Author Response · Authors · 2024-11-20
> > **Author Response Part 2**
> >
> > >**[W2] A detailed comparison with TabSyn**
> >
> > **Main Difference: Data Space vs. Latent Space.** TabSyn is a latent diffusion model that uses a VAE to project data of different types into homogeneous latent codes before applying modeling the generative process with diffusion. In contrast, TabDiff is a hybrid diffusion model that directly operates in the data space. It designs customized diffusion processes tailored to best fit the nature of each data type.
> >
> >
> > This distinction makes TabDiff a more natural and direct approach to modeling mixed-type tabular data, as it avoids potential information loss during the VAE modeling.
> >
> > **Training Efficiency and Stability.** The distinction between latent and hybrid diffusion models has a direct impact on the training process. The latent model TabSyn requires a two-phase training process: first, training a VAE to obtain latent codes, and then training the diffusion model. On the contrary, TabDiff trains directly on data in a single phase.
> >
> > Furthermore, as mentioned in [1], the quality of the VAE has a significant impact on TabSyn’s performance, as it's the only component that recovers the original data shape. We encountered this issue when reproducing TabSyn’s results.
> >
> > We observed that across different training runs, the sample quality varies significantly. For poorly performing runs, we attempted to retrain the diffusion model and even increased the number of training epochs, but these efforts did not improve the results. This confirms that the issue lies with the VAE.
> >
> > On the contrary, TabDiff consistently maintains robust performance across different checkpoints from different training runs.
> >
> > To further illustrate this, we present the results of Shape and Trend that are averaged across 10 random training runs in the following table (Note: in the paper, we follow the convention of previous works and reported results based on 20 different runs of the same checkpoint, and we put the result of the best reproduction run for TabSyn). These additional results demonstrate that TabDiff achieves significantly more consistent performance across different training runs, as shown by the smaller average and standard deviation.
> >
> > | **Method**        | **Adult**              | **Default**            | **Beijing**            |
> > |-|-|-|-|
> > | TabSyn (Shape) | 1.31 $\pm$ **0.64**    | **1.17** $\pm$ **0.21**    | 2.69 $\pm$ **1.44**    |
> > | TabSyn (Trend) | 2.73 $\pm$ **0.98**    | 5.05 $\pm$ **2.22**    | 5.05 $\pm$ **1.88**    |
> > | TabDiff (Shape) | **0.65** $\pm$ 0.08    | 1.19 $\pm$ 0.12    | **1.07** $\pm$ 0.6     |
> > | TabDiff (Trend) | **1.47** $\pm$ 0.18    | **2.46** $\pm$ 0.62    | **2.61** $\pm$ 0.20    |
> >
> >
> > **Learnable noise schedules help dependency modeling.** Another significant difference between the models is that TabDiff adopts adaptive learnable schedules that allow features to be generated at different rates. Whereas, TabSyn has the same noise schedule for all features (VAE-projected features to be more precise). The learnable schedules promote the network to learn to generate different features in flexible orders, thus enhancing the modeling of feature-wise correlations. Such enhancement is empirically verified by TabDiff’s consistently superior performance on the column-wise correlation metrics, as shown in Table 2.
> >
> > [1] Zhang et al., 2024 “Mixed-Type Tabular Data Synthesis with Score-based Diffusion in Latent Space.”
> >
> > ---
> >
> > Hope our response can address your questions!

---

> ### Author Response · Authors · 2024-11-24
> **Looking forward to feedback during the discussion period**
>
> Dear Reviewer gKXC,
>
> We would like to express our gratitude for your valuable comments and suggestions. Thanks a lot for your positive feedback on our method and experiments.
>
> As the author-reviewer discussion period **is coming to a close on Nov 27th**, we would greatly appreciate it if you could clarify any remaining concerns you may have. This will ensure that we can adequately address them during this period.
>
> Thank you very much for your attention to this matter.
>
> Best regards,
>
> Authors

---

> > ### Comment · Reviewer_gKXC · 2024-11-25
> >
> > I want to thank the authors for their response. I agree that denoising network conditions can enhance multi-modal modeling. Your response, along with the comparison between data space and latent space, has addressed my initial confusion. I will maintain my score.

---

> > > ### Author Response · Authors · 2024-11-27
> > > **Thank you very much for your feedback!**
> > >
> > > Thank you very much for providing valuable feedback and recognizing our efforts!

---

### Public Comment · ~Markus_Mueller1 · 2024-11-25

Dear authors,

Thank you for this very interesting read and your contribution to diffusion models for tabular data. It is great to see progress on noise schedules specifically designed for tabular data.

We have a specific interest in your paper as it is closely related to our work on diffusion models for tabular data [1]. In our paper, we cast both categorical and continuous features into continuous space, to be able to apply Gaussian noise with adaptive noise schedules to both feature types.

Below we list a couple of questions that would help us (and hopefully you as well) to better understand the impact of some modeling choices:

- We found that our modeling choices help to balance model capacity allocation across features. Did you also find a similar balancing impact of using feature-specific noise schedules in your work? Would the balancing across features be sensitive to how different the scales of numerical and categorical losses are?
- In your work, the categorical features are generated via unmasking and cannot change afterwards. In that case, have you seen any deterioration in results if categorical features depend a lot on the fine-grained details of the continuous features (which may be generated only after a categorical feature has already been unmasked)?
- Your choice of letting $\lambda_{\text{num}}$ linearly decay as training proceeds is interesting. Essentially this seems to imply that towards the end of training parameter updates will be devoted towards modeling categorical features. In our own experiments, however, we found that modeling categorical features is rather easy compared to getting the fine details of continuous features right. What is the rationale behind your choice?

---

[1] Mueller, et al. (2024) "Continuous Diffusion for Mixed-Type Tabular Data".

---

> ### Author Response · Authors · 2024-11-27
> **Thank you for the related work!**
>
> Hi Markus,
>
> Thank you for bringing the related work to our attention. We are delighted to discuss our specific modeling choices and believe that this discussion will deepen understanding for both you and us!
>
>
> > **[Q1] Regarding the balancing impact of feature-specific noise schedules**
>
> Yes, we do observe such an impact. As detailed in Appendix F.2, TabDiff with feature-specific schedules demonstrates more uniform column-wise error distributions compared to baseline approaches.
>
> Regarding how the different scales of continuous and discrete loss impact this balancing effect: this is a very good consideration. The different scales in loss could indeed have an effect. But, in our experiments, we haven’t observed how unbalanced loss impacts the balancing effect. This could be attributed to the fact that we thoroughly trained both modalities thoroughly with designs such as the loss calibration techniques as discussed in Q3. However, we do consider that a more thorough study of loss scales’ impact would be valuable for future work, particularly in extreme cases where the loss scales differ by several orders of magnitude.
>
> > **[Q2] The case when a categorical feature depends on the fine-grained details of some continuous features**
>
> This is an insightful question. First, we want to clarify that TabDiff’s sampling process, as outlined in Algorithm 2, does allow unmasked categorical features to potentially change their values in later sampling steps. In line 6 of Algorithm 2, our stochastic sampler first stochastically perturbs the current sample to a slightly higher noise level, before making the prediction. Thus, an unmasked categorical feature might become masked again and get re-predicted at a later timestep, when the fine-grained details of the continuous features are better built.
>
> Even if the stochastic sampler is disabled, we haven't observed performance degradation in such cases in our experiment. This might be attributed to the fact that feature dependencies are typically flexible and arbitrary, with no specific order. If categorical features depend on the fine-graded details of continuous features, then the reverse is also true. Given this, our generation process can accommodate such a case: when a categorical feature is unmasked early in the generation process, the continuous features and other unmasked categorical features would need to adjust their own content based on the fine-grained details provided by the unmasked feature.
>
> > **[Q3] Decaying continuous feature loss weight**
>
> Yes, your rationale is exactly correct. In our experiment, we observed that under our current loss scale, the discrete loss takes a bit longer to get thoroughly trained than the continuous. We speculate this opposite phenomenon, relative to your finding, might be due to the difference in the how model's categorical features or the difference in our loss scales.
>
> Given our observation that the training loss has a direct correlation with sample quality, we would want a technique that thoroughly trained the discrete feature while avoiding overfitting the continuous features. A linearly decreasing weight loss turns out to be an effective and robust solution that generalizes well across all datasets.
>
> ---
> We hope this addresses your questions!
>
> Authors

---

> > ### Public Comment · ~Markus_Mueller1 · 2024-11-27
> >
> > Dear Authors,
> >
> > thank you very much for answering my questions.
> >
> > Best of luck for the rest of the discussion period!
> >
> > Markus

---

### Meta-Review · Area_Chair_3QbP · 2024-12-12

**Metareview:**

This paper explores “multi-modal” diffusion generative models for tabular data, where “multi-modal” refers to numerical and categorical tabular data only. The work develops a joint continuous-time process with learned columnwise noise schedules that help to model the heterogeneous features typical in tabular data.

Reviewers initially had a range of concerns, but these were generally addressed during the discussion.

While I am recommending acceptance as a poster, I have two requirements that the authors must address in their camera-ready version:

1) Change the title and writing throughout the paper to deemphasize “multi-modal”, as this term is not being used in a standard sense. The paper only considers the tabular modality, which can have mixed-type columns (numerical and categorical). Hence changing “multi-modal data” to “mixed-type tabular data” for example would better communicate the scope of the work.

2) Reduce the length of the main text to fit within the ICLR page limit.

Authors, please post a comment on this thread once you have made these changes and I will review. If I do not see your post and changes by the camera-ready deadline I will raise the issue to SAC/PC. Thank you for cooperating!

**Additional Comments On Reviewer Discussion:**

The main points of improvement raised by reviewers were: confusion about the term “multi-modal”; relationship to prior/concurrent work including TabSyn, TabDDPM, and CDTD; questions on how numerical and categorical data are jointly modelled; some lack of details, explanations, and motivations throughout; lack of computational costs comparison; and other minor points.

While this is a long list of issues, they were generally addressed by the authors and satisfied the reviewers. Several reviewers increased their scores in response, and all reviewers agreed on acceptance.

---

> ### Public Comment · ~Minkai_Xu1 · 2025-02-16
>
> Dear AC,
>
> Thank you for your feedback and we have addressed your two requirements accordingly for our camera-ready version.
> Please let us know if there are other concerns!
>
> Sincerely,
> Authors

---

> > ### Comment · Area_Chair_3QbP · 2025-02-16
> >
> > Thank you for the update and taking the reviewer's and my own feedback into account.
> >
> > AC

---

### Decision · Program_Chairs · 2025-01-22

Accept (Poster)